# Improving Task-Specific Multimodal Sentiment Analysis with General MLLMs via Prompting

**Haoyu Zhang**[†], **Yinan Zhang**[†], **Chaolong Ying**[†], **Xiaoying Tang**[‡], **Tianshu Yu**[†*]

[†]School of Data Science, The Chinese University of Hong Kong, Shenzhen
[‡]School of Science and Engineering, The Chiese University of Hong Kong, Shenzhen
{haoyuzhang3,yinanzhang1,chaolongying}@link.cuhk.edu.cn
{tangxiaoying,yutianshu}@cuhk.edu.cn

## Abstract

Multimodal Sentiment Analysis (MSA) aims to predict sentiment from diverse data types, such as video, audio, and language. Recent progress in Multimodal Large Language Models (MLLMs) have demonstrated impressive performance across various tasks. However, in MSA, the increase in computational costs does not always correspond to a significant improvement in performance, raising concerns about the cost-effectiveness of applying MLLMs to MSA. This paper introduces the MLLM-Guided Multimodal Sentiment Learning Framework (MMSLF). It improves the performance of task-specific MSA models by leveraging the generalized knowledge of MLLMs through a teacher-student framework, rather than directly using MLLMs for sentiment prediction. First, the proposed teacher built upon a powerful MLLM (e.g., GPT-4o-mini), guides the student model to align multimodal representations through MLLM-generated context-aware prompts. Then, knowledge distillation enables the student to mimic the teacher's predictions, thus allowing it to predict sentiment independently without relying on the context-aware prompts. Extensive experiments on the SIMS, MOSI, and MOSEI datasets demonstrate that our framework enables task-specific models to achieve state-of-the-art performance across most metrics. This also provides new insights into the application of general MLLMs for improving MSA.[1]

## 1 Introduction

Multimodal Sentiment Analysis (MSA) aims to predict sentiment from various types of input, such as language, video, and audio. Accurate MSA is crucial for several applications, such as Human-Computer Interaction and Healthcare [1, 2]. Compared to unimodal sentiment analysis, the mutually complementary nature of multiple modalities typically leads to better performance, thereby improving the applicability of MSA in real-world scenarios.

A series of studies focused on improving MSA through well-designed representational learning and multimodal fusion networks. For example, Tsai et al. [3] introduces a novel model which employs multiple Transformers for pairwise alignment of modality information. Hazarika et al. [4] propose a method to disentangle each modality into modality-invariant and modality-specific features for multi-perspectives fusion. Additionally, Yu et al. [5] apply self-supervised learning to generate pseudo-labels for each modality to learn both modality consistency and inconsistency. Zhang et al. [6] make language modality as dominant modality to guide the learning of representations in other modalities, thus mitigating potential conflicts between different modalities. After years of exploration,

---

[*]Correspondence author
[1]Code: https://github.com/LOGO-CUHKSZ/MMSLF

39th Conference on Neural Information Processing Systems (NeurIPS 2025).

it has become increasingly challenging to achieve performance improvement in MSA. Fortunately, recent multimodal large language models (MLLMS) have demonstrated notable performance for various specific tasks [7–11, 2]. For example, Lian et al. [2] explores the application of GPT-4V [12] for MSA, showing that MLLMs without finetuning can achieve performance comparable to many task-specific models through their general knowledge. However, the parameters of task-specific models mostly range from several million to tens of millions. Compared to these task-specific models, the increased parameter count and computational costs of general MLLMs does not always lead to a significant improvement in performance, raising concerns about the MLLMs' cost-effectiveness. This inspired us to explore whether it is possible to apply general MLLMs knowledge to assist in the training of task-specific MSA models, thus achieving better MSA.

In this paper, we aim to bridge the gap between task-specific models and MLLMs in MSA by leveraging the generalized knowledge of MLLMs to help with training task-specific models. To this end, we introduce the MLLM-Guided Multimodal Sentiment Learning Framework (MMSLF), which embeds an MLLM within the teacher network to provide enhanced supervision for the task-specific student model, thereby avoiding the direct use of the MLLMs for sentiment prediction. In the teacher network, we use a pre-trained MLLM (*e.g.,* GPT-4o-mini [12]) to generate context-aware prompts that highlight key sentiment cues across different modalities. These prompts guide the model to learn conditional attention maps in specially designed alignment modules, helping it better capture sentiment information. The student network is a task-specific model that learns from the guidance of the teacher. It receives the same multimodal inputs but does not use prompts from MLLMs. Instead, it aligns the sentiment information from conditional attention and features learned by the teacher to improve performance of sentiment analysis. Extensive experiments on popular datasets, such as SIMS [13], MOSI [14], and MOSEI [15] demonstrate the effectiveness of MMSLF, showing its state-of-the-art performance. In summary, our work makes the following contributions, which introduce a novel solution to the challenges in MSA:

- We explore using the general knowledge of MLLMs to guide the training of task-specific MSA models, offering new insight into applying general MLLMs to improve MSA.

- We design a conditional alignment mechanism that enables the teacher model with MLLM's knowledge to intuitively and efficiently guide the student model's multimodal alignment and representation learning.

- Extensive comparisons and ablation studies on three popular datasets (*e.g.,* SIMS, MOSI, and MOSEI) demonstrate that the proposed MMSLF can improve the training process of task-specific models, enabling them to achieve state-of-the-art performance across most metrics.

## 2 Related Work

### 2.1 Multimodal Sentiment Analysis

Multimodal Sentiment Analysis (MSA) aims to predict human sentiment by leveraging various types of data, such as video, audio, and text. Early methods, such as TFN [16] and LMF [17], achieved state-of-the-art performance by capturing relationships between modalities through Cartesian product-based tensor fusion. However, these methods face the challenge of rapidly increasing computational costs as the feature dimensions and the number of modalities grow. With the advent of deep learning architectures, the attention mechanism has become popular in the design of MSA methods [3, 18, 4, 19–22, 6]. For example, MulT [3] employs multi-head attention to align modalities, facilitating more effective multimodal fusion. ALMT [6] leverages language representations at different scales to guide the learning of other auxiliary modalities, mitigating the influence of noise that can negatively impact fusion. In addition, various other novel methods [23, 5, 24] have also made significant progress in the MSA. For example, Yu et al. [5] proposed generating uni-modal sentiment labels to help the model capture both consistency and differentiation across modalities. Moreover, Yuan et al. [24] introduced an adversarial training strategy based on semantic reconstruction using original-noisy instance pairs, achieving robust MSA in simulated noisy scenarios. Despite these progress, achieving further improvements in performance remains challenging. A recent study [2] explored the application of GPT-4V in MSA, demonstrating that MLLMs can achieve performance comparable to small-scale models. Different from this work, our work utilizes MLLMs to help the learning of task-specific models rather than directly using MLLMs for MSA.

## 2.2 Large Language Models

In recent years, large language models (LLMs) have made remarkable strides, with models such as GPT-3 [25], T5 [26], and LLaMA [27] demonstrating impressive capabilities by scaling both data and model sizes. However, despite these advances, uni-modal LLMs are limited to processing text-based information, restricting their applicability to a broader range of tasks and scenarios. To overcome this limitation, researchers have explored the potential of multimodal large language models (MLLMs), building upon the foundation of uni-modal LLMs. Significant progress has been made in developing powerful MLLMs [28–31, 8–10, 32–35], showcasing their surprising practical capabilities. For instance, GPT-4V [12] integrates natural language processing with visual understanding to analyze images and provide textual responses to questions about them. Similarly, LLaVA [7] translates visual content into text by employing a linear layer to embed images, making the LLMs understand visual input. Video-LLaMA [8] achieving multimodal understanding by aggregating representations from different modalities after applying positional embedding through Q-formers [32]. Moreover, Zhao et al. [10] introduced MMICL, which leverages multimodal in-context learning to achieve state-of-the-art performance on various visual language tasks. In this work, we utilize MLLMs to generate prompts for smaller task-specific models, enabling efficient multimodal learning.

## 2.3 Teacher-Student Models

The teacher-student framework has been widely applied in knowledge distillation, particularly for knowledge compression [36]. It focuses on transferring knowledge from a larger teacher model to a smaller student model through carefully designed strategies, such as soft label matching [37–40] and feature matching [41–44]. For example, Hinton et al. [37] introduced the use of the teacher model's probability distribution as soft labels to guide the student model's learning process. By utilizing these soft labels, the student model is trained not only to predict the correct labels but also to closely align with the teacher model's soft predictions, thereby facilitating effective knowledge transfer. Additionally, Zagoruyko et al. [43] proposed an attention transfer method that improves the student model's performance by transferring activation-based and gradient-based attention maps from the teacher model. In the context of MSA, recent advancements include MC-Teacher [40], which introduced learnable pseudo-label selection and self-adaptive exponential moving average strategies to achieve semi-supervised MSA. In this work, we employ feature matching and attention transfer techniques to achieve our research objectives. To the best of our knowledge, this is the first attempt to transfer the general knowledge of MLLMs to smaller models for MSA.

# 3 Method

## 3.1 Overview

The overall pipeline of the MMSLF is illustrated in Figure 1. First, with the given preprocessed multimodal input sequences, each modality is processed through three embedding layers. Then, the extracted features are aligned using a designed Conditional Alignment module, where the condition is provided by prompts from MLLMs (*e.g.,* GPT-4o-mini). Specifically, visual and audio features are aligned with language features via two alignment modules: Visual-to-Language (V $\rightarrow$ L) Alignment and Audio-to-Language (A $\rightarrow$ L) Alignment. These conditional alignment layers establish correspondences between modalities with the help of the MLLM's prompt, facilitating effective multimodal alignment. Finally, the multimodal fusion module combines the aligned features to produce a unified representation, which is used to predict the final sentiment score via a regression loss $L_{\text{regr}}^{\text{Teacher}}$ (defined as Eq. 9).

Once the teacher is trained, the student is trained to mimic the behavior of the teacher. The key difference between the student and teacher is that the student align video and audio features with language features directly, without the conditional input (*i.e.,* MLLM's Prompt) used in the teacher. Additionally, instead of using the regression loss of sentiment scores $L_{\text{regr}}^{\text{Student}}$ (defined as Eq. 12), two regularization techniques are used to help the student learn from the teacher: (1) the student's attention maps are trained to match the teacher's conditional attention maps using an attention transfer loss $\mathcal{L}_{\text{attn}}^{\text{Student}}$ (defined as Eq. 10), and (2) the fused unified representations of the student are encouraged to match those of the teacher through a unified representation matching loss $\mathcal{L}_{\text{fusion}}^{\text{Student}}$ (defined as Eq. 11). These loss ensure that the model captures the same patterns as the teacher.

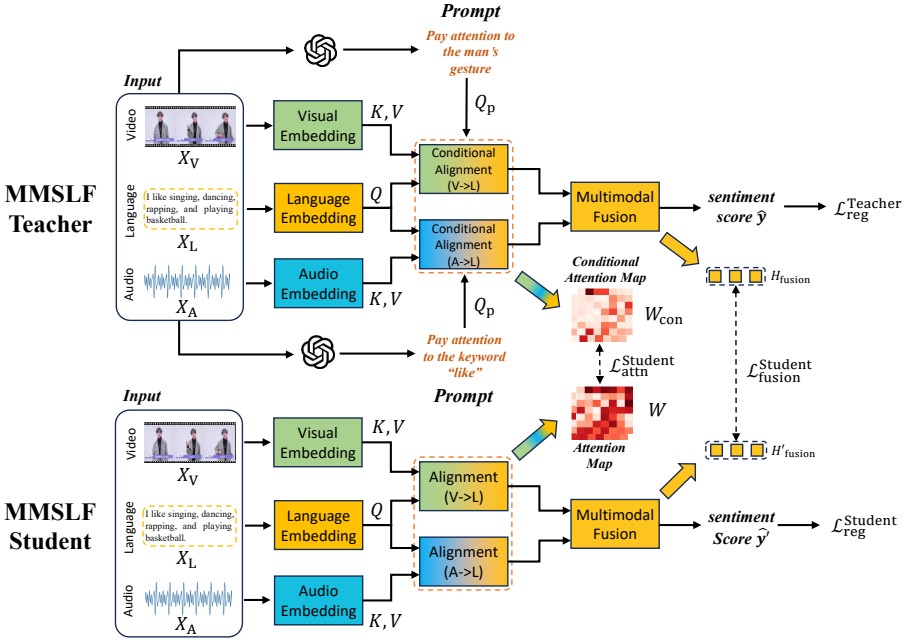

Figure 1: Overall pipeline of MMSLF. Note: L, A, and V refer to language, audio, and visual modalities, respectively.

## 3.2 Multimodal Input and Embedding

We utilize the preprocessed sequences in the datasets as inputs. Specifically, the language input is processed using BERT [45], while visual input is handled by OpenFace [46], and audio input is processed with Librosa [47]. We denote the multimodal input as $X_m \in \mathbb{R}^{T_m \times d_m}$, where $m \in \{L, A, V\}$, $T_m$ represents the length of the input sequence, and $d_m$ indicates the vector dimension.

Given the multimodal input $X_m$, we apply three embedding layers $\mathrm{E}_m$, each consisting of a linear layer to extract features from each modality and map them into a unified feature dimension $d$:

$$S_m = \mathrm{E}_m(X_m; \theta_{\mathrm{E}_m}) \in \mathbb{R}^{T_m \times d}, \tag{1}$$

where $S_m$ represents the embedded features of modality $m$, and $\theta_{\mathrm{E}_m}$ denotes the parameters associated with each embedding layer.

## 3.3 Multimodal Alignment

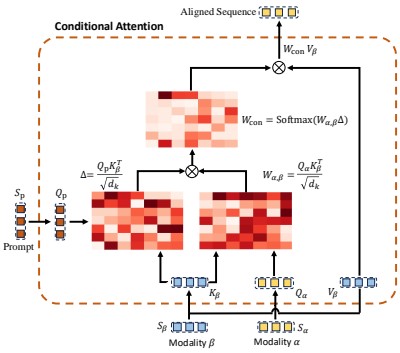

Figure 2: An example of conditional attention used to align modality $\beta$ to modality $\alpha$.

**Prompt Embedding.** To extract features from the MLLMs' prompt $X_\mathrm{P}$ and fix the feature dimension to $d$, we apply a pre-trained BERT along with an embedding layer (comprising a Transformer encoder with a depth of two layers) to $X_\mathrm{P}$. We denote the combined operation as $\mathrm{E}_\mathrm{P}$. The process can be described as:

$$S_\mathrm{P} = \mathrm{E}_\mathrm{P}(X_\mathrm{P}; \theta_{\mathrm{E}_\mathrm{P}}) \in \mathbb{R}^{T_\mathrm{L} \times d}, \tag{2}$$

where $S_\mathrm{P}$ represents the embedded feature of the prompt, which has the same feature shape as $S_\mathrm{L}$, and $\theta_{\mathrm{E}_\mathrm{P}}$ denotes the parameters used in the MLLMs, pre-trained BERT, and the embedding layer. In practice, for the V->L alignment, $X_\mathrm{P}$ contains the prompt information from both visual and language modalities. For the A->L alignment, $X_\mathrm{P}$

contains the prompt information from the audio and language modalities. However, since GPT-4o-mini does not support audio analysis, we only include language information with A->L alignment. We also experimented with generating prompts using Gemini-2.0-Flash, which supports audio input, but its performance was not better than GPT-4o-mini. For detailed discussions, please refer to Section 4.7 and Section 4.8.

**Conditional Attention.** To introduce the general knowledge of MLLMs for assistance in model training, we added conditional inputs based on the multi-head attention mechanism. As illustrated in Figure 2, to align modality $\beta$ to modality $\alpha$, the module first uses $S_\alpha$ to compute Query ($Q_\alpha$), while $S_\beta$ is used to compute the Key ($K_\beta$) and Value ($V_\beta$). The relationship/attention map $W_{\alpha,\beta}$ between these two modalities is computed as follows:

$$W_{\alpha,\beta} = \frac{Q_\alpha K_\beta^{\mathrm{T}}}{\sqrt{d_k}} \in \mathbb{R}^{T_\alpha \times T_\beta}, \tag{3}$$

where $d_k$ denotes the dimension of each attention head, and $T_\alpha$ and $T_\beta$ represent the sequence lengths of the corresponding modalities. Simultaneously, we apply the prompt $S_{\mathrm{P}}$ as a conditional Query ($Q_{\mathrm{P}}$) to $K_\beta$ and $V_\beta$ to compute a shifted attention map $\Delta \in \mathbb{R}^{T_\alpha \times T_\beta}$. Then, we obtained the conditional attention map $W_{con}$ by fusing $W_{\alpha,\beta}$ and $\Delta$:

$$W_{\mathrm{con}} = \mathrm{softmax}(W_{\alpha,\beta} \cdot \Delta) \in \mathbb{R}^{T_\alpha \times T_\beta}, \tag{4}$$

where the $\mathrm{softmax}$ represents weight normalization operation. Finally, the aligned feature $H_{\beta\to\alpha}^{\mathrm{Teacher}}$ can be computed as follows:

$$H_{\beta\to\alpha}^{\mathrm{Teacher}} = \text{Feed-Forward}(W_{\mathrm{con}}V_\beta; \theta_{\mathrm{fwd}}) \in \mathbb{R}^{T_\alpha \times d}, \tag{5}$$

where Feed-Forward and $\theta_{\mathrm{fwd}}$ represent the MLPs and corresponding parameters. In practice, the conditional attention layer is used to replace the original attention layer in the Transformer decoder [48, 3] while keeping the other components unchanged.

**Conditional Alignment in Teacher.** The teacher aligned the obtained $S_{\mathrm{V}}$ and $S_{\mathrm{A}}$ to $S_{\mathrm{L}}$ using the designed Conditional Alignment module. Specifically, the MLLMs' prompts is used to specify which sentiment cues in each modality require more attention, thus helping the teacher better capture aligned sentiment information across these modalities. We denote the aligned outputs as $H_{\mathrm{V}\to\mathrm{L}}^{\mathrm{Teacher}}$ and $H_{\mathrm{A}\to\mathrm{L}}^{\mathrm{Teacher}}$ which are then utilized for multimodal fusion. For example, the process that align visual modality to language modality can be described as:

$$H_{\mathrm{V}\to\mathrm{L}}^{\mathrm{Teacher}} = \mathrm{CondlAlignment}(X_{\mathrm{V}}, X_{\mathrm{L}} \mid X_{\mathrm{P}}; \theta_{\mathrm{V}\to\mathrm{L}}^{\mathrm{Teacher}}) \in \mathbb{R}^{T_{\mathrm{L}} \times d}, \tag{6}$$

where $\mathrm{CondAlignment}$ represents the Conditional Alignment module, $X_{\mathrm{P}}$ denotes the prompt from MLLMs, $\theta_{\mathrm{V}\to\mathrm{L}}^{\mathrm{Teacher}}$ is the parameters used to align the modalities.

**Alignment in Student.** The alignment module in student is designed to learn the relationships between modalities independently (*i.e.,* learning without the help of MLLMs' prompts). We denote the outputs of the module as $H_{\mathrm{V}\to\mathrm{L}}^{\mathrm{Student}}$ and $H_{\mathrm{A}\to\mathrm{L}}^{\mathrm{Student}}$. For example, the $H_{\mathrm{V}\to\mathrm{L}}^{\mathrm{Student}}$ can be obtained by:

$$H_{\mathrm{V}\to\mathrm{L}}^{\mathrm{Student}} = \mathrm{Alignment}(X_{\mathrm{V}}, X_{\mathrm{L}}; \theta_{\mathrm{V}\to\mathrm{L}}^{\mathrm{Student}}) \in \mathbb{R}^{T_{\mathrm{L}} \times d}, \tag{7}$$

where $\mathrm{Alignment}$ and $\theta_{\mathrm{V}\to\mathrm{L}}^{\mathrm{Student}}$ represent the Alignment module and parameters, respectively.

## 3.4 Multimodal Fusion and Prediction

With these features extracted from the various modalities, we employ a Transformer encoder with self-attention blocks for multimodal fusion. In practice, we concatenate the obtained features with a randomly initialized and learnable regression token $H_{\mathrm{fusion}} \in \mathbb{R}^{1 \times d}$ as input, then the Transformer encoder can transfer and compress essential information to the $H_{\mathrm{fusion}}$, thus making sentiment prediction through this token. For the final sentiment prediction, we apply a linear layer to $H_{\mathrm{fusion}}$:

$$\hat{y} = \mathrm{Regression}(H_{\mathrm{fusion}}; \theta_{\mathrm{regr}}) \in \mathbb{R}^1, \tag{8}$$

where $\hat{y}$ denotes the predicted sentiment score, $\mathrm{Regression}$ represents the linear layer, and $\theta_{\mathrm{regr}}$ represents the parameters of the linear layer.

## 3.5 Learning Objectives

As outlined in Section 3.1, the training of MMSLF consists of two stages: (1) training the teacher and (2) training the student. In the first stage, the teacher learns to perform MSA under the guidance of prompts from MLLMs. The overall learning objective is defined as:

$$\mathcal{L}_{\text{overall}}^{\text{Teacher}} = \mathcal{L}_{\text{regr}}^{\text{Teacher}} = \frac{1}{N} \sum_{i=1}^{N} |\hat{y}^i - y^i|, \tag{9}$$

where $N$ is the number of samples in the training set, $y^i$ is the sentiment label of the $i$-th sample, $\hat{y}^i$ is the prediction of teacher. In the second stage, the student is trained under the supervision of the teacher, whose parameters remain frozen. The attention transfer loss $\mathcal{L}_{\text{attn}}^{\text{Student}}$ is formulated as:

$$\mathcal{L}_{\text{attn}}^{\text{Student}} = \frac{1}{N} \sum_{i=1}^{N} |W^i - W_{\text{con}}^i|, \tag{10}$$

where $W^i$ is the attention map from the last layer of the alignment module in the student, and $W_{\text{con}}^i$ is the conditional attention map from the last layer of the conditional alignment module in teacher. The fused unified representation matching loss $\mathcal{L}_{\text{fusion}}^{\text{Student}}$ is defined as:

$$\mathcal{L}_{\text{fusion}}^{\text{Student}} = \frac{1}{N} \sum_{i=1}^{N} |H_{\text{fusion}}'^i - H_{\text{fusion}}^i| \tag{11}$$

where $H_{\text{fusion}}'^i$ and $H_{\text{fusion}}^i$ represent the fused features from the student and teacher, respectively. The sentiment prediction loss for the student is defined as:

$$\mathcal{L}_{\text{regr}}^{\text{Student}} = \frac{1}{N} \sum_{i=1}^{N} |\hat{y'}^i - y^i|, \tag{12}$$

where $\hat{y'}^i$ is the prediction of student. Overall, the learning objective of student is:

$$\mathcal{L}_{\text{overall}}^{\text{Student}} = \mathcal{L}_{\text{regr}}^{\text{Student}} + \alpha \mathcal{L}_{\text{attn}}^{\text{Student}} + \beta \mathcal{L}_{\text{fusion}}^{\text{Student}}, \tag{13}$$

where the $\alpha$ and $\beta$ are empirically chosen hyperparameters. In practice, for the SIMS dataset, $\alpha$ and $\beta$ are set to 60.0 and 8.0, respectively, while for the MOSI dataset, they are set to 100.0 and 4.0. For more discussion of the hyperparameters, please refer to Appendix C.1 and Appendix B.9.

# 4 Experiment and Analysis

## 4.1 Datasets

**SIMS.** SIMS [13] is a Chinese MSA dataset, with data sourced from Chinese movies, TV series, and variety shows, featuring complex real-world scenarios. It consists of 1,368 training samples, 456 validation samples, and 457 test samples. Each sample is annotated with a continuous sentiment score ranging from -1 to 1, where -1 represents negative sentiment, and 1 represents positive sentiment.

**MOSI.** MOSI [14] is an English MSA dataset, composed of data collected from YouTube. The dataset includes 1,284 training samples, 229 validation samples, and 686 test samples. Each instance is manually annotated with a continuous sentiment score ranging from -3 to 3, with -3 representing strongly negative and 3 representing strongly positive.

**MOSEI.** MOSEI [15] is an English MSA dataset with data collected from YouTube. It contains 22,856 video clips, including 16,326 training samples, 1,871 validation samples, and 4,659 test samples. Similar to MOSI, each sample is manually annotated with a score ranging from -3 to 3.

## 4.2 Baselines

We compare our method with several advanced task-specific MSA methods, whose model parameters range from several million to tens of millions. These methods include: TFN [16], LMF [17], MuLT [3], MISA [4], Self-MM [5], TETFN [21], ALMT [6], DLF [49], and concurrent work DeepMLF [50].

The performance of these models is all reproduced using a popular framework MMSA [51]. We also include some MLLMs, such as Video-LLaMA2 [9], GPT-4V [12], GPT-4o-mini [12], and Gemini-2.0-Flash [52], for comparison. Additionaly, due to factors including differences in experimental settings, the lack of open-source implementation for certain methods, and space limitations, we have conducted additional comparisons in Appendix B.1 for more detailed comparison and discussion.

## 4.3 Evaluation Criteria

Consistent with previous studies [4, 6], we evaluate the regression tasks by reporting the mean absolute error (MAE) and the correlation between the model's predictions and human annotations (Corr). Since the predicted sentiment score can be used to compute classification accuracy, we also report Acc-2 and F1 scores for all datasets. For example, scores $> 0$ are treated as positive while scores $\leq 0$ are non-positive. This method is widely used in the MSA studies. Additionally, in line with prior work [4, 6], we report accuracy based on both negative/positive and negative/non-negative classifications for the MOSI and MOSEI datasets. In the tables, performance metrics computed using these two classification methods are separated by a "/", with the left side representing negative/non-negative performance and the right side representing negative/positive performance. All results are averaged over five runs, with standard deviations reported.

Since there are significant differences in the performance of many MLLMs between classification tasks and regression tasks, directly calculating classification accuracy based on regression metrics leads to poor performance for models like GPT-4o-mini. To more accurately demonstrate the true capabilities of MLLMs, we conducted two evaluations on the MLLMs. One of the tests involved using a classification prompt template to evaluate the Acc-2 and F1 metrics, while the other involved using a regression prompt template to assess the MAE and Corr metrics. For a fair comparison, all task-specific models were evaluated in the same manner, *i.e.,* using Acc-2 (SIMS)/negative/non-negative Acc-2 (MOSI and MOSEI) to determine the model parameters for evaluation of classification performance and using MAE to determine the model parameters for evaluation of regression performance. More related discussions can be found in Limitation.

## 4.4 Performance Comparison

Table 1, Table 2 and Table 3 present the performance results on SIMS, MOSI and MOSEI, respectively. Gemini-2.0-Flash, which is a advanced MLLM at present, performs the best in most metrics. Notably, the performance of the teacher is close to the GPT-4o-mini in many metrics, and it outperforms both Video-LLaMA2 and GPT-4V in all metrics on all datasets. Furthermore, compared to Video-LLaMA2 and GPT-4V, both the teacher and student demonstrate improvements across most metrics. For example, on the SIMS, the student achieves an Acc-2 of 81.40±1.58, marking a relative improvement of 1.64% over Video-LLaMA2. When compared to the task-specific model ALMT, student achieves a 2.10% relative improvement in F1 on the SIMS. A similar trend is observed on the MOSI and MOSEI dataset (Table 2), showing the general applicability of MMSLF across cultures, *i.e.,* both Chinese and English datasets. Moreover, it is worth noting that the student can achieve advanced performance with fewer parameters compared to MLLMs, which underscores the potential of task-specific models in the MSA field. Furthermore, as shown in the Table 3, the results on the larger dataset MOSEI show that teacher/-Student achieves advanced performance on many metrics. This demonstrates that MMSLF has good generalization ability on datasets with different sizes. It is worth noting that the concurrent work DeepMLF demonstrates notable performance across the three datasets. By introducing a task-specific MLLM with learnable tokens for multimodal fusion, DeepMLF brings a promising research direction for MSA. Alongside MMSLF, it further highlights the potential of MLLMs in advancing the MSA field.

## 4.5 Effect of Each Component

In Table 4, we show the results by removing specific components. First, when we removed the MLLMs' prompt from the teacher, we observed a significant drop in performance across both datasets. Specifically, on the SIMS dataset, the F1 score decreased from 84.06% to 80.84%, and MAE increased from 0.370 to 0.436. A similar trend was observed on the MOSI dataset, where the F1 score dropped from 85.15% to 79.60%, and MAE increased from 0.734 to 0.914. These phenomenoa show that the MLLMs plays a crucial role in helping the model capture relevant multimodal information

Table 1: Performance comparison on SIMS dataset. $a$ represents the results reproduced by the authors from open-source code with default hyperparameters. $b$ represents the results are from [2]. $c$ represents the results are from [13].

| Method | Acc-2 | F1 | MAE | Corr |
|---|---|---|---|---|
| Video-LLaMA2[a] | 80.09 | 79.94 | 0.584 | 0.476 |
| GPT-4V[b] | 81.24 | - | - | - |
| GPT-4o-mini[a] | 82.71 | 82.51 | 0.453 | 0.663 |
| Gemini-2.0-flash[a] | **85.12** | **84.69** | **0.381** | **0.747** |
| TFN[a] | 78.12±1.56 | 77.83±1.62 | 0.434±1.12 | 0.579±1.50 |
| MISA[a] | 77.72±1.10 | 76.54±1.67 | 0.451±1.83 | 0.570±1.95 |
| Self-MM[a] | 77.94±1.11 | 77.72±0.68 | 0.418±1.05 | 0.589±1.54 |
| TETFN[a] | 80.18±0.49 | 79.34±0.52 | 0.422±1.30 | 0.588±1.71 |
| ALMT[a] | 79.91±0.29 | 80.17±0.60 | 0.421±0.69 | 0.583±0.70 |
| DeepMLF[a] | 82.89±2.37 | 83.09±2.32 | 0.362±0.30 | 0.720±0.30 |
| **MMSLF** | | | | |
| *Teacher* | **83.06±0.95** | **84.06±0.43** | **0.370±0.50** | **0.690±0.80** |
| *Student* | 81.40±1.58 | 81.85±1.41 | 0.382±1.39 | 0.662±1.26 |

Table 2: Performance comparison on MOSI dataset. $a$ represents the results reproduced by the authors from open-source code with default hyperparameters, while $b$ represents the results are from [2].

| Method | Acc-2 | F1 | MAE | Corr |
|---|---|---|---|---|
| Video-LLaMA2[a] | 83.24/86.43 | 82.60/86.23 | 1.149 | 0.696 |
| GPT-4V[b] | 80.43/- | - | - | - |
| GPT-4o-mini[a] | 87.32/89.48 | 87.17/89.42 | 0.997 | 0.842 |
| Gemini-2.0-flash[a] | **87.76/89.49** | **87.74/91.61** | **0.633** | **0.856** |
| TFN[a] | 77.38±1.37/78.11±0.60 | 77.35±1.33/78.02±0.57 | 0.949±3.13 | 0.662±1.95 |
| MISA[a] | 80.93±0.99/81.05±0.83 | 80.90±1.03/81.01±0.87 | 0.773±1.81 | 0.775±0.63 |
| Self-MM[a] | 82.94±0.63/83.18±0.35 | 82.95±0.63/83.09±0.36 | 0.717±1.53 | 0.792±0.55 |
| TETFN[a] | 80.87±0.52/80.82±0.53 | 80.87±0.52/80.82±0.53 | 0.726±1.68 | 0.791±0.86 |
| ALMT[a] | 83.00±0.22/85.12±0.20 | 83.00±0.22/85.19±0.27 | **0.713±0.75** | 0.795±0.54 |
| DLF[a] | -/83.69±0.29 | -/83.71±0.27 | 0.761±1.81 | 0.781±0.48 |
| DeepMLF | 79.65±1.04/81.10±0.85 | 79.52±1.17/81.04±0.95 | 0.795±2.41 | 0.758±1.12 |
| **MMSLF** | | | | |
| *Teacher* | **85.05±0.66/86.61±0.69** | **85.15±0.66/86.69±0.69** | 0.734±1.46 | **0.797±0.60** |
| *Student* | 83.62±0.91/85.37±1.00 | 83.68±0.96/85.50±0.96 | 0.746±1.63 | 0.775±1.10 |

Table 3: Performance comparison on MOSEI dataset. $a$ represents the results reproduced by the authors from open-source code with default hyperparameters.

| Method | Acc-2 | F1 | MAE | Corr |
|---|---|---|---|---|
| Video-LLaMA2[a] | 83.29/84.50 | 83.23/85.21 | 0.922 | 0.406 |
| GPT-4o-mini[a] | 85.04/**86.90** | 85.25/**87.04** | 1.015 | **0.744** |
| Gemini-2.0-flash[a] | **88.07**/58.86 | **88.08**/72.08 | **0.583** | 0.743 |
| TFN[a] | 83.00±0.45/82.90±0.43 | 82.68±0.40/82.83±0.41 | 0.566±0.31 | 0.725±0.21 |
| MISA[a] | 84.41±0.30/85.09±0.62 | 84.16±0.30/85.02±0.59 | 0.553±0.46 | 0.759±0.25 |
| Self-MM[a] | 84.15±0.50/84.90±0.49 | 84.15±0.43/84.79±0.40 | **0.529±0.47** | 0.764±0.45 |
| TETFN[a] | 84.18±0.62/85.42±0.43 | 84.06±0.63/85.31±0.55 | 0.543±0.51 | 0.769±0.27 |
| ALMT[a] | 84.35±0.34/84.76±0.45 | 84.10±0.32/84.25±0.59 | 0.542±0.45 | 0.768±0.17 |
| DLF[a] | -/84.76±0.32 | -/84.70±0.35 | 0.543±0.11 | 0.759±0.30 |
| DeepMLF | 83.49±0.52/**86.67±0.40** | 83.79±0.45/86.57±0.41 | 0.510±0.42 | 0.800±0.24 |
| **MMSLF** | | | | |
| *Teacher* | **85.08±0.36/86.62±0.75** | **85.55±0.24/86.71±0.71** | 0.539±1.06 | **0.773±1.51** |
| *Student* | 83.96±0.38/84.67±0.27 | 84.20±0.48/84.74±0.28 | 0.548±0.41 | 0.747±0.51 |

more effectively. Second, we removed the guidance of the teacher during the training of the student. This led to a decrease in the student model's performance, with the F1 score on SIMS dropping from 81.85% to 78.72%, and on MOSI from 83.68% to 83.00%. The increase in MAE values on both datasets also reflects the student model's reduced ability to align multimodal information without teacher guidance. It also shows that the importance of knowledge distillation, as the teacher's guidance can help the student learn the relationship between each modality effectively. Furthermore, the ablation results on the MOSEI dataset can be found in Appendix B.10.

Table 4: Effect of each component.

| Method | SIMS | | MOSI | |
|---|---|---|---|---|
| | F1 | MAE | F1 | MAE |
| **MMSLF-Teacher** | **84.06±0.43** | **0.370±0.50** | **85.15±0.66/86.69±0.69** | **0.734±1.46** |
| *w/o prompt* | 80.84±0.93 | 0.436±0.57 | 79.60±0.95/81.21±1.07 | 0.914±0.68 |
| **MMSLF-Student** | **81.85±1.41** | **0.382±1.39** | **83.68±0.96/85.50±0.96** | **0.746±1.63** |
| *w/o guid. of teacher* | 78.72±0.53 | 0.429±1.02 | 83.00±0.59/85.07±0.52 | 0.743±1.30 |

## 4.6 Effect of Each Regularization

To evaluate the effect of each regularization in the student, we removed $\mathcal{L}_{attn}^{Student}$, $\mathcal{L}_{fusion}^{Student}$, and both $\mathcal{L}_{fusion}^{Student}$ and $\mathcal{L}_{attn}^{Student}$. The results are presented in Table 5. We observe that both F1 and MAE decrease when each regularization is removed, indicating that every regularization contributes positively to the performance of student. Moreover, it is evident that the impact of each regularization is more significant on the SIMS dataset than on the MOSI dataset. For example, when $\mathcal{L}_{attn}^{Student}$ is removed, the F1 score drops by a relative 3.24% on SIMS, while it decreases by only 1.11% on MOSI. These differences could be attributed to the varying levels of difficulty between the MOSI datasets. Additionally, we tried different combinations of $\alpha$ and $\beta$, please see Appendix B.9 for more details.

Table 5: Effect of each regularization.

| Method | SIMS | | MOSI | |
|---|---|---|---|---|
| | F1 | MAE | F1 | MAE |
| **MMSLF-Student** | **81.85±1.41** | **0.382±1.39** | **83.68±0.96/85.50±0.96** | **0.746±1.63** |
| *w/o $\mathcal{L}_{attn}^{Student}$* | 79.28±0.75 | 0.453±0.48 | 82.76±0.30/84.80±0.42 | 0.741±0.71 |
| *w/o $\mathcal{L}_{fusion}^{Student}$* | 79.23±0.69 | 0.428±0.87 | 83.16±0.51/85.44±0.55 | 0.738±0.76 |
| *w/o $\mathcal{L}_{fusion}^{Student}$ & $\mathcal{L}_{attn}^{Student}$* | 78.72±0.53 | 0.429±1.02 | 83.00±0.59/85.07±0.52 | 0.743±1.30 |

## 4.7 Effect of Prompt from Different MLLMs

To evaluate the impact of different prompts on task-specific model performance, we compared the results of using prompts generated by Gemini-2.0-Flash and GPT-4o-mini. As shown in Table 6, the teacher and student models guided by GPT-4o-mini outperform those guided by Gemini-2.0-Flash across most metrics. Despite Gemini-2.0-Flash's additional capability to process audio, its prompts are less effective, resulting in a significant performance decrease (further discussion on the quality of prompts can be found in Section 4.8). This indicates that the quality of prompts generated by large models is important to the performance of task-specific models, and that optimizing prompt generation for specific tasks can significantly improve task-specific models' performance.

## 4.8 Analysis of Sentiment Cues in MLLMs' Prompt

To further analyze the impact of prompts generated by different large models, we show the results of sentiment classification using prompts from different modalities in the Table 7. The results indicate that Gemini-2.0-Flash performs significantly worse than GPT-4o-mini across multiple metrics. For example, in the SIMS dataset, Gemini-2.0-Flash has an Acc-2 and F1 of 18.82% and 31.68%, respectively, for the linguistic cue, while GPT-4o-mini achieves 78.99% and 79.64%. A similar trend is observed in the MOSI dataset. These results indicate that, despite Gemini-2.0-Flash having the

Table 6: Effect of prompt from different MLLMs.

| Method (MLLM) | SIMS | | | |
|---|---|---|---|---|
| | Acc-2 | F1 | MAE | Corr |
| **MMSLF (Gemini-2.0-Flash)** | | | | |
| *Teacher* | 81.09±0.23 | 81.09±0.29 | 0.377±0.73 | 0.686±1.53 |
| *Student* | 80.00±0.41 | 80.11±0.54 | 0.422±0.96 | 0.627±1.70 |
| **MMSLF (GPT-4o-mini)** | | | | |
| *Teacher* | **83.06±0.95** | **84.06±0.43** | **0.370±0.50** | **0.690±0.80** |
| *Student* | 81.40±1.58 | 81.85±1.41 | 0.382±1.39 | 0.662±1.26 |

| Method (MLLM) | MOSI | | | |
|---|---|---|---|---|
| | Acc-2 | F1 | MAE | Corr |
| **MMSLF (Gemini-2.0-Flash)** | | | | |
| *Teacher* | 80.58±0.67/82.59±0.53 | 80.51±0.69/82.59±0.55 | 0.865±2.55 | 0.711±2.57 |
| *Student* | 83.56±0.38/85.37±0.54 | 83.49±0.37/85.26±0.54 | **0.722±0.93** | 0.783±0.56 |
| **MMSLF (GPT-4o-mini)** | | | | |
| *Teacher* | **85.05±0.66/86.61±0.69** | **85.15±0.66/86.69±0.69** | 0.734±1.46 | **0.797±0.60** |
| *Student* | 83.62±0.91/85.37±1.00 | 83.68±0.96/85.50±0.96 | 0.746±1.63 | 0.775±1.10 |

added capability of analyzing the audio modality compared to GPT-4o-mini, the accuracy of the prompts it generates is still not as good as that of GPT-4o-mini. They may contain more misleading information, which may result in poorer performance when using prompts generated by Gemini-2.0-Flash for model training as discussed in Section 4.7. This also indicates that the accuracy of the sentiment information in the prompts has a significant impact on the task-specific model.

Table 7: Analysis of sentiment cues in MLLMs' prompt. The left side and right side of "/" are Acc-2 and F1, respectively.

| Prompts Source | SIMS | | | MOSI | | |
|---|---|---|---|---|---|---|
| | L | V | A | L | V | A |
| Gemini-2.0-Flash | 18.82/31.68 | 14.22/24.90 | **4.60/8.79** | 83.39/83.37 | 57.29/50.35 | **56.85/49.91** |
| GPT-4o-mini | **78.99/79.64** | **78.56/75.82** | - | **86.88/86.87** | **76.68/75.84** | - |

## 5 Conclusion and Future Work

In this paper, we explored the application of MLLMs in MSA tasks and assumed that MLLMs can assist task-specific models during training, thus achieving better performance. To validate this idea, we introduced a novel MLLM-guided Multimodal Sentiment Learning Framework (MMSLF). This framework leverages general-purpose MLLMs to generate prompts that guide the learning process of a teacher model. The teacher subsequently transfers the acquired knowledge to a student model, enabling it to perform inference independently without further reliance on MLLM-generated prompts. Extensive experiments on the SIMS, MOSI, and MOSEI datasets demonstrate that MMSLF achieves competitive performance across most metrics, thereby confirming our initial assumption and offering new insights into the integration of MLLMs for MSA tasks.

Moreover, current MSA research faces two key challenges. First, achieving significant performance improvements has become increasingly difficult. One of the reasons is that many current research over-relies on conventional feature extraction toolkits such as Librosa and OpenFace. While effective, these tools may not provide sufficiently rich representations for advanced architectures. Exploring more powerful pretrained multimodal feature extractors could offer new directions for progress. Second, reproducibility remains a critical issue. Many existing MSA datasets are relatively small, leading to high training variance and sensitivity to initialization. This data scarcity not only hinders model generalization but also complicates fair comparisons across studies. Therefore, we argue that there is a need to establish larger, more diverse, and well-annotated MSA benchmarks. In our future work, we aim to address these limitations and extend the proposed framework to broader affective computing tasks.

## Acknowledgement

This work was supported in part by the National Science and Technology Major Project of China (2022ZD0116408) and by the Guangdong Provincial Key Laboratory of Mathematical Foundations for Artificial Intelligence (2023B1212010001).

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

# A Limitations

In this paper, we explored the possibility of using prompts from general MLLMs to help model alignment, thus improving model's performance. To effectively verify the feasibility of the ideas, we use simple and intuitive model designs. However, this also leads to some limitations of the model. (1) Due to the use of traditional feature extractors, the performance of MMLSF has not been fully realized. For example, in Appendix B.3, the model's performance improved after replacing with a more powerful feature extractor. (2) Due to the large number of parameters of MMSLF-Teacher, when training it on small-scale datasets such as MOSI and SIMS, the phenomenon of overfitting is extremely severe. (3) Due to the small parameters of the MMSLF-Student, the model training is unstable (as shown in Appendix B.11). During the training process, we found that there were significant differences in performance under different random seeds.

# B Additional Experiments and Analysis

## B.1 Additional Comparison

Despite the differences of the settings, such as five runs, in the MSA method, we still to use the original results to ensure a comprehensive comparison. This methods includes TFN [16], LMF [17], MuLT [3], MAG-BERT [18], MISA [4], Self-MM [5], TETFN [21], CENET [22], ALMT [6], KuDA [53], DMD [54], MMML [55] (See Appendix B.3), DLF [49], TCAN [56] and concurrent work DeepMLF [50].

As illustrated in Table 8, Table 9, and Table 10, we use MAE as the key metric to determine the model checkpoints which is consistent with the comparative methods. It is obvious MMSLF demonstrates competitive performance across all datasets, with particularly strong results on the student model. We also observed an interesting phenomenon: the student slightly outperforms the teacher on some metrics. For example, the teacher achieves an Acc-2 of 76.71 while the student achieves an Acc-2 of 80.18. We believe that this occurs because we retained direct supervision from the ground truth labels rather than fully aligning with the teacher during student training, so some of the teacher's subtle biases were not transferred. These phenomenon demonstrate the effectiveness of our proposed framework. In addition, we can see that DeepMLF achieves excellent performance on multiple datasets, which is inseparably linked to its architecture design that directly utilizes LLMs such as GPT-2.

Table 8: Performance comparison on SIMS dataset. $a$ represents the results are from MMSA [51]. $b$ represents the results reproduced by the authors from open-source code.

| Method | Acc-2 | F1 | MAE | Corr |
|---|---|---|---|---|
| KuDA | 80.74 | 80.71 | 0.408 | 0.613 |
| TFN[a] | 78.38 | 78.62 | 0.432 | 0.591 |
| LMF[a] | 77.77 | 77.88 | 0.441 | 0.576 |
| MulT[a] | 78.56 | 79.66 | 0.453 | 0.564 |
| MISA[a] | 76.54 | 76.59 | 0.447 | 0.563 |
| Self-MM[a] | 80.04 | 80.44 | 0.425 | 0.595 |
| TETFN[a] | 81.18 | 80.24 | 0.420 | 0.577 |
| CENET[a] | 77.90 | 77.53 | 0.470 | 0.540 |
| ALMT[b] | 78.08±0.63 | 78.24±0.58 | 0.421±0.69 | 0.583±0.70 |
| DeepMLF[b] | **82.19±0.80** | **82.50±0.74** | **0.362±0.30** | **0.720±0.30** |
| **MMSLF** | | | | |
| *Teacher* | 76.71±3.05 | 77.31±2.72 | 0.370±0.50 | 0.690±0.80 |
| *Student* | 80.18±1.07 | 80.18±1.08 | 0.382±1.39 | 0.662±1.26 |

## B.2 Generality of the Proposed Framework

To evaluate the generality of the MMSLF, we applied the Teacher-Student framework to ALMT [6]. As shown in Table 11, ALMT-Teacher outperformed MMSLF-Teacher across all metrics on both the MOSI datasets, demonstrating the effectiveness of utilizing MLLMs to improve the learning of

Table 9: Performance comparison on MOSI dataset. $a$ represents the results are from MMSA [51]. $b$ represents the results reproduced by the authors from open-source code.

| Method | Acc-2 | F1 | MAE | Corr |
|---|---|---|---|---|
| MAG-BERT | -/86.10 | -/86.00 | 0.712 | 0.796 |
| DMD | -/83.23 | -/83.29 | 0.752 | - |
| TCAN | -/86.28 | -/86.15 | 0.714 | 0.797 |
| KuDA | 84.40/86.43 | 84.48/86.46 | **0.705** | 0.795 |
| TFN[a] | 77.99/79.08 | 77.95/79.11 | 0.947 | 0.673 |
| LMF[a] | 77.9/79.18 | 77.8/79.15 | 0.950 | 0.651 |
| MuLT[a] | 79.71/80.98 | 79.63/80.95 | 0.880 | 0.702 |
| MISA[a] | 81.84/83.54 | 81.82/83.58 | 0.777 | 0.778 |
| Self-MM[a] | 83.44/83.36 | 85.46/85.43 | 0.708 | 0.796 |
| TETFN[a] | 83.24/85.37 | 83.13/85.33 | 0.708 | **0.798** |
| CENET[a] | 83.53/85.21 | 83.49/85.22 | 0.725 | 0.795 |
| ALMT[b] | 82.22±0.83/84.12±0.55 | 82.15±0.87/84.11±0.55 | 0.713±0.75 | 0.795±0.54 |
| DLF[b] | -/83.66±0.44 | -/83.70±0.43 | 0.761±1.81 | 0.781±0.48 |
| DeepMLF[b] | 81.60±0.74/83.08±0.84 | 81.55±0.78/83.08±0.88 | 0.795±2.41 | 0.758±1.12 |
| **MMSLF** | | | | |
| *Teacher* | **84.32±0.92/85.89±1.07** | **84.20±0.96/85.82±1.10** | 0.734±1.46 | 0.797±0.60 |
| *Student* | 82.52±0.42/84.14±0.75 | 83.49±0.51/84.18±0.74 | 0.746±1.63 | 0.775±1.10 |

Table 10: Performance comparison on MOSEI dataset. $a$ represents the results are from MMSA [51]. $b$ represents the results reproduced by the authors from open-source code.

| Method | Acc-2 | F1 | MAE | Corr |
|---|---|---|---|---|
| DMD | -/84.62 | -/84.62 | 0.543 | - |
| TCAN | -/86.27 | -/86.17 | 0.532 | 0.774 |
| KuDA | 83.26/86.46 | 82.97/86.59 | 0.529 | 0.776 |
| TFN[a] | 78.50/81.89 | 78.96/81.74 | 0.573 | 0.714 |
| LMF[a] | 80.54/80.94 | 83.48/83.36 | 0.576 | 0.717 |
| MulT[a] | 81.15/84.63 | 81.56/84.52 | 0.559 | 0.733 |
| MISA[a] | 80.67/84.67 | 81.12/84.66 | 0.558 | 0.752 |
| Self-MM[a] | 83.76/85.15 | 83.82/84.90 | 0.531 | 0.765 |
| TETFN[a] | **84.12**/86.21 | **84.35**/86.11 | 0.537 | 0.770 |
| CENET[a] | 83.52/**86.38** | 83.85/86.32 | 0.526 | 0.778 |
| ALMT[b] | 83.28±0.40/85.16±0.39 | 83.20±0.78/85.14±0.76 | 0.542±0.45 | 0.768±0.17 |
| DLF[b] | -/84.53±0.52 | -/84.49±0.47 | 0.543±0.11 | 0.759±0.30 |
| DeepMLF[b] | 81.57±0.71/86.23±0.30 | 82.16±0.60/**86.27±0.27** | **0.510±0.42** | **0.800±0.24** |
| **MMSLF** | | | | |
| *Teacher* | 83.55±1.61/85.55±0.73 | 83.74±1.23/85.31±0.90 | 0.539±1.06 | 0.773±1.51 |
| *Student* | 81.44±2.21/85.13±0.44 | 82.00±2.02/85.09±0.29 | 0.520±0.33 | 0.741±0.60 |

task-specific small models. However, ALMT-Student did not exhibit the same level of improvement as MMSLF-Student. We attribute this isbecause that ALMT was not originally designed with the Teacher-Student framework. Its reliance on multiple specialized attention maps complicates the optimization of the student model during the knowledge distillation process. Additionally, it is worth noting that MMSLF-Student achieved better results than ALMT-Student with a significantly smaller number of parameters, further demonstrating the effectiveness and efficiency of the MMSLF.

## B.3 Modality Extractor Analysis

Some recent methods like DeepMLF [50] and MMML [55] use advanced feature extractors to help the model to achieve better performance. We also investigated the impact of the feature extractor on the MMSLF's performance and compared it with MMML. Consistent with MMML, we ues RoBERTa and Data2Vec as the text and audio modality extractor, respectively. Moreover, we used MAE to determine the model parameters and report the results with three runs. The results are shown in the Table 12 below. We can see that MMSLF achieve performance improvement using more powerful feature extractors. Compare with MMML, although MMML achieves higher performance

Table 11: Generality of the proposed framework.

| Method | Acc-2 | F1 | MAE | Corr |
|---|---|---|---|---|
| | | SIMS | | |
| **ALMT** | | | | |
| *Teacher* | **84.20±0.57** | **84.45±0.81** | **0.363±0.76** | **0.711±1.50** |
| *Student* | 79.87±1.81 | 80.58±1.05 | 0.418±2.15 | 0.587±3.97 |
| **MMSLF** | | | | |
| *Teacher* | 83.06±0.95 | 84.06±0.43 | 0.370±0.50 | 0.690±0.80 |
| *Student* | 81.40±1.58 | 81.85±1.41 | 0.382±1.39 | 0.662±1.26 |
| | | MOSI | | |
| **ALMT** | | | | |
| *Teacher* | **86.56±0.68/88.02±0.67** | **86.63±0.69/88.06±0.68** | **0.677±0.57** | **0.834±0.46** |
| *Student* | 83.26±0.41/85.43±0.14 | 83.38±0.31/85.52±0.15 | 0.720±0.54 | 0.784±0.28 |
| **MMSLF** | | | | |
| *Teacher* | 85.05±0.66/86.61±0.69 | 85.15±0.66/86.69±0.69 | 0.734±1.46 | 0.797±0.60 |
| *Student* | 83.62±0.91/85.37±1.00 | 83.68±0.96/85.50±0.96 | 0.746±1.63 | 0.775±1.10 |

on Acc2-Has0 (86.32%) and F1-Has0 (86.23%), our method outperforms on more metrics. For example, the teacher achieves better results on Corr (0.792±0.15). The student achieves better results on Acc2-Non0 (87.09±0.25), F1-Non0 (87.18±0.24) and MAE (0.513±1.27). These results indicate that it is feasible to use updated and more powerful extractors to achieve better MSA performance.

Table 12: Comparison with MMML on MOSEI dataset.

| Method | Acc-2 | F1 | MAE | Corr |
|---|---|---|---|---|
| MMML | **86.32**/86.73 | **86.23**/86.49 | 0.517 | 0.791 |
| MMSLF-Teacher | 85.47±0.25 / 87.06±0.46 | 85.53±0.27 / 87.16±0.38 | 0.522±1.31 | **0.792±0.15** |
| MMSLF-Student | 85.93±0.58 / **87.09±0.25** | 86.08±0.55 / **87.18±0.24** | **0.513±1.27** | 0.785±1.66 |

## B.4 Efficiency-Performance Trade-off Analysis

As shown in Table 13, we quantitatively presented the efficiency-performance trade-off analysis in terms of parameters, GFLOPs, and inference time on SIMS datasets. Note that the reported parameter counts do not include those of the feature extractors used for serialization, such as OpenFace [46], Librosa [47], and BERT [45]. This is because tools like Librosa lack well-defined parameter counts. More importantly, excluding them allows for a fairer comparison focused on the core modeling components across methods. The overall computational complexity of the models can be assessed using the GFLOPs metric.

Obviously, our student model achieves competitive performance (F1 of 81.85±1.41 on SIMS) with only 0.82M parameters, 8.6 GFLOPs, and 6.39s test-time inference. This demonstrate that our methods can achieve better efficiency-performance trade-off than other methods. In contrast, Gemini-2.0-Flash and GPT-4o-mini require larger parameters and >22min inference time. These results demonstrate that our method offers a trade-off between performance and computational cost. In addition, it is also worth noting that although MMSLF has a small number of parameters, its GFLOPs are relatively higher. This is because we do not compress the input sequences' length as the prior methods [4, 6]. With further optimization (*e.g.,* sequence dimension reduction), the computational cost of MMSLF can be further reduced.

## B.5 Significance Analysis

As shown in Table 14 below, we show two-tailed t-tests between the Student model and the task-specific baseline ALMT. The resulting p-values are MAE=0.0029, Corr=0.00022, Acc-2=0.095, and F1=0.066. Under the conventional 0.05 threshold, the Student's improvements on MAE and Corr are statistically significant, while the gains on ACC-2 and F1 show the same positive trend.

Table 13: Comparison of efficiency and performance on the SIMS dataset.

| Method | Parameters | GFLOPs | Inference Time | F1 | MAE |
|---|---|---|---|---|---|
| GPT-4V | > 7B | - | > 30min | 81.24 | - |
| GPT-4o-mini | > 7B | - | > 27min | 82.51 | 0.453 |
| Gemini2.0-Flash | > 7B | - | > 22min | **84.69** | 0.381 |
| TFN | 35.63M | **0.101** | **3.46s** | 77.83±1.62 | 0.434±1.12 |
| MISA | 21.66M | 7.33 | 12.32s | 76.54±1.67 | 0.451±1.83 |
| Self-MM | **0.38M** | 6.66 | 11.40s | 77.72±0.68 | 0.418±1.05 |
| TETFN | 1.53M | 6.72 | 26.57s | 79.34±0.52 | 0.422±1.30 |
| ALMT | 2.60M | 7.00 | 16.08s | 80.17±0.60 | 0.421±0.69 |
| MMSLF-Teacher | 2.54M | 96.16 | > 12.31s + 27min | 84.06±0.43 | **0.370±0.50** |
| MMSLF-Student | 0.82M | 8.61 | 6.39s | 81.85±1.41 | 0.382±1.39 |

Table 14: Two-tailed t-tests between the Student model and the ALMT on SIMS dataset.

| Method | p-value (MAE) | p-value (Corr) | p-value(Acc.2) | p-value (F1) |
|---|---|---|---|---|
| MMSLF-Student & ALMT | 0.0029 | 0.00022 | 0.095 | 0.066 |

## B.6 Prompt Sampling Analysis

Considering both cost and performance, we obtain all prompts once before training the teacher model. We also experimented with generating three prompts per sample and randomly sampling one during training on SIMS dataset, but found that this way lead to higher cost without significant performance improvement. As shown in Table 15 below, the "No Sampling" setting achieves comparable or even better results. Moreover, we observe that the "Sampling" strategy introduces noticeably higher variance for MMSLF-Teacher, especially in MAE and Corr (*e.g.,* 2.71 vs. 0.50 for MAE std), suggesting that sampling different prompts may introduce inconsistent guidance and lead to unstable training. However, for MMSLF-Student, the variance across runs is relatively small in both settings (*e.g.,* F1 std: 1.41 vs. 0.66), indicating that the student model is less sensitive to prompt sampling. This is likely because the student learns from the teacher's distilled representations and attention patterns, rather than directly using the prompts.

Table 15: Effect of prompt sampling methods on performance.

| Method | Acc-2 | F1 | MAE | Corr |
|---|---|---|---|---|
| MMSLF-Teacher (Sampling) | 83.57±1.73 | 83.23±1.30 | 0.370±2.71 | 0.682±4.69 |
| MMSLF-Student (Sampling) | 81.05±0.66 | 81.18±0.66 | 0.385±0.92 | 0.667±0.87 |
| MMSLF-Teacher (No Sampling) | 83.06±0.95 | 84.06±0.43 | 0.370±0.50 | 0.690±0.80 |
| MMSLF-Student (No Sampling) | 81.40±1.58 | 81.85±1.41 | 0.382±1.39 | 0.662±1.26 |

## B.7 Prompt Sensitivity Analysis

As shown in Table 16 below, we evaluated our method using different prompt templates to assess robustness. The experimental results in the table show small performance variance across different prompt formulations within the same MLLM. This indicate that advanced large-scale MLLMs may generate stable outputs regardless of reasonable prompt variations. This stability validates our framework's robustness to prompt engineering choices. In addition, the performance difference between GPT-4o-mini and Gemini-2.0-Flash may stem from their varying abilities to generate accurate and informative prompts for sentiment-centered multimodal tasks, rather than sensitivity to prompt variations. Therefore, compared to changing the prompt input, the more important is to select a appropriate mLLM with strong domain-specific capabilities.

## B.8 Performance Impact of Varying Student Parameters

Table 17 presents the performance impact of different parameter settings on the student model. We control the model parameters by modifying the depth of the alignment module. Notably, the student

Table 16: Comparison of Prompt Sensitivity and Robustness on SIMS datasets.

| Method | Prompt | Acc-2 | F1 | MAE | Corr |
|---|---|---|---|---|---|
| Teacher | Prompt 1 (Default, GPT-4o-mini) | **83.06±0.95** | **84.06±0.43** | 0.370±0.50 | 0.690±0.80 |
| Teacher | Prompt 2 (GPT-4o-mini) | 81.23±5.94 | 83.93±1.16 | **0.346±1.12** | **0.716±1.47** |
| Teacher | Prompt 3 (GPT-4o-mini) | 80.04±5.37 | 82.51±0.71 | 0.372±1.20 | 0.687±2.72 |
| Teacher | Prompt 4 (Gemini-2.0-Flash) | 81.09±0.23 | 81.09±0.29 | 0.377±0.73 | 0.686±1.53 |
| Teacher | Prompt 5 (Gemini-2.0-Flash) | 77.90±7.03 | 82.87±1.21 | 0.355±2.25 | 0.704±3.08 |
| Teacher | Prompt 6 (Gemini-2.0-Flash) | 71.42±4.11 | 81.95±0.07 | 0.384±1.47 | 0.658±1.82 |
| Student | Prompt 1 (Default, GPT-4o-mini) | **81.40±1.58** | **81.85±1.41** | 0.382±1.39 | 0.662±1.26 |
| Student | Prompt 2 (GPT-4o-mini) | **81.40±1.09** | 81.65±0.96 | 0.394±1.06 | 0.667±0.45 |
| Student | Prompt 3 (GPT-4o-mini) | 80.74±0.37 | 80.92±0.74 | 0.393±1.80 | 0.664±1.34 |
| Student | Prompt 4 (Gemini-2.0-Flash) | 80.00±0.41 | 80.11±0.54 | 0.422±0.96 | 0.627±1.70 |
| Student | Prompt 5 (Gemini-2.0-Flash) | 81.18±0.97 | 81.15±0.89 | 0.387±1.92 | 0.670±2.69 |
| Student | Prompt 6 (Gemini-2.0-Flash) | 81.05±0.45 | 81.14±0.37 | **0.381±0.96** | **0.676±1.59** |

achieves optimal performance with 0.82M parameters, corresponding to a configuration (as shown in Table 20) of 1 embedding layers, 2 alignment layers, and 2 multimodal fusion layers. beyond this point, increasing the model size does not significantly improve the performance, showing that the model has likely already fully utilized its learning capacity.

Table 17: Performance Comparison of Varying Student Model Parameters on SIMS dataset. The parameters from BERT used for input preprocessing are excluded from the reported parameter count.

| Method | Parm. | Acc-2 | F1 | MAE | Corr |
|---|---|---|---|---|---|
| ALMT | 2.60M | 79.91±0.29 | 80.17±0.60 | 0.421±0.69 | 0.583±0.70 |
| Teacher | 2.54M | **83.06±0.95** | **84.06±0.43** | **0.370±0.50** | **0.690±0.80** |
| | 0.49M | 80.74±1.16 | 81.44±1.03 | 0.408±1.52 | 0.638±2.15 |
| | 0.82M | **81.40±1.58** | 81.85±1.41 | **0.382±1.39** | **0.662±1.26** |
| Student | 1.46M | 80.66±0.51 | 81.47±0.54 | 0.400±1.64 | 0.631±1.72 |
| | 2.11M | 81.36±1.29 | **82.32±0.75** | 0.394±1.33 | 0.646±1.43 |
| | 4.05M | **81.40±0.71** | 81.79±0.50 | 0.394±1.65 | 0.636±1.93 |

## B.9 Effect of Regularization Weight on Model Performance

To investigate the impact of regularization weights, we experimented with various combinations of $\alpha$ and $\beta$ on the SIMS dataset. The results are presented in Table 18. It is evident that both $\alpha$ and $\beta$ influence the performance of the student.

Table 18: Effect of regularization weight on model performance

| $\alpha$ | $\beta$ | Acc-2 | F1 | MAE | Corr |
|---|---|---|---|---|---|
| 60.0 | 8.0 | **81.40±1.58** | **81.85±1.41** | **0.382±1.39** | **0.662±1.26** |
| 80.0 | 8.0 | 81.01±1.51 | 81.27±1.34 | 0.394±1.40 | 0.650±2.36 |
| 40.0 | 8.0 | 81.18±1.66 | 81.44±1.52 | 0.388±1.15 | **0.662±1.58** |
| 20.0 | 8.0 | 80.79±1.29 | 81.46±1.17 | 0.387±1.33 | 0.661±1.53 |
| 0 | 8.0 | 77.94±1.12 | 79.28±0.75 | 0.453±0.48 | 0.524±1.87 |
| 60.0 | 10.0 | 81.01±1.87 | 81.27±1.67 | 0.389±1.20 | 0.656±1.30 |
| 60.0 | 6.0 | 80.88±1.26 | 81.37±0.92 | 0.392±1.53 | 0.653±1.82 |
| 60.0 | 4.0 | 80.74±1.01 | 81.23±1.16 | 0.393±1.63 | 0.650±2.26 |
| 60.0 | 2.0 | 80.53±0.97 | 81.05±0.99 | 0.396±1.06 | 0.645±2.29 |
| 60.0 | 0 | 78.29±0.42 | 79.23±0.69 | 0.428±0.87 | 0.564±3.10 |
| 0 | 0 | 78.56±0.44 | 78.72±0.53 | 0.429±1.02 | 0.567±1.39 |

## B.10  Supplement of Each Component Effect

As shown in Table 19, we present the ablation experiments of each component on the MOSEI dataset as a supplement to Section 4.5. We can see that each module is still useful on the MOSEI dataset, demonstrating the effectiveness of the MMSLF.

Table 19: Effect of each component on MOSEI dataset.

| Method | MOSEI | |
|---|---|---|
| | F1 | MAE |
| **MMSLF-Teacher** | **85.55±0.24/86.71±0.71** | **0.539±0.71** |
| *w/o prompt* | 84.11±0.78/85.07±0.48 | 0.558±0.78 |
| **MMSLF-Student** | **84.20±0.48/84.74±0.28** | **0.548±0.41** |
| *w/o guid. of teacher* | 83.34±0.52/84.83±0.21 | 0.553±0.34 |

## B.11  Convergence Performance Analysis

In Figure 3, we visualize the loss curves of student on the SIMS and MOSI datasets. While the overall trend shows a decrease, the variance of $\mathcal{L}_{\text{attn}}^{\text{Student}}$ across different seeds is relatively high. We believe this is due to the difficulty student faces in aligning with the teacher's learning outcomes without the help of MLLMs' prompts, resulting in fluctuations during the optimization process. Despite this, student still achieves competitive performance on both the SIMS and MOSI datasets, demonstrating the effectiveness of the proposed MMSLF framework.

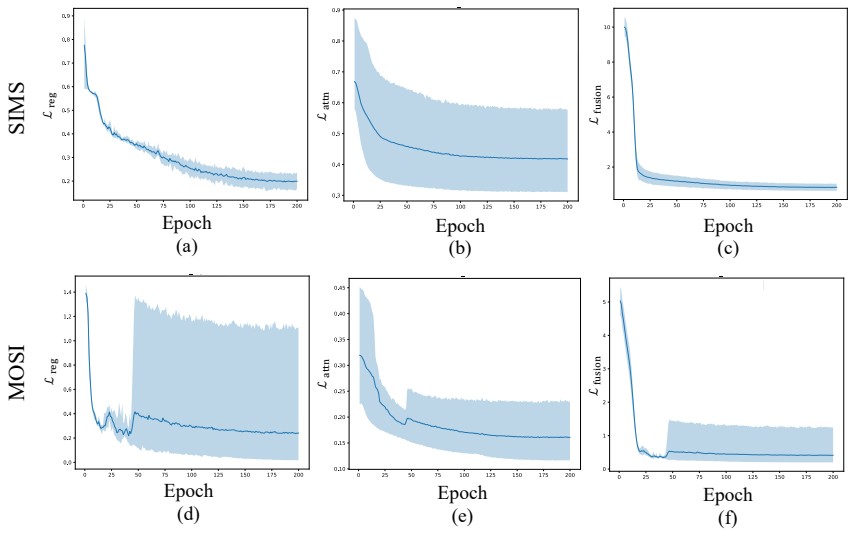

Figure 3: Visualization of convergence curves. The values on the curve represent the averages corresponding to five runs.

## B.12  Case Study of Conditional Attention Map

As shown in Figure 4, we visualized the attention difference maps by subtracting the attention map without MLLMs' prompts from the conditional attention map $H_{\text{V}\rightarrow\text{L}}^{\text{Teacher}}$. In combination with the key cue from large model, although these frames look similar, the first frame shows more pronounced mouth movements, while the third frame displays more obvious muscle movements in both the eyes and mouth. It is also important to note that the yellow frames merely indicate that the model assigns them lower attention under the guidance of the MLLM's prompts, not that they lack sentiment information. They are just relatively less important compared to the frames highlighted in blue. The observations indicate that the MLLM's prompts is useful for task-specific models' learning.

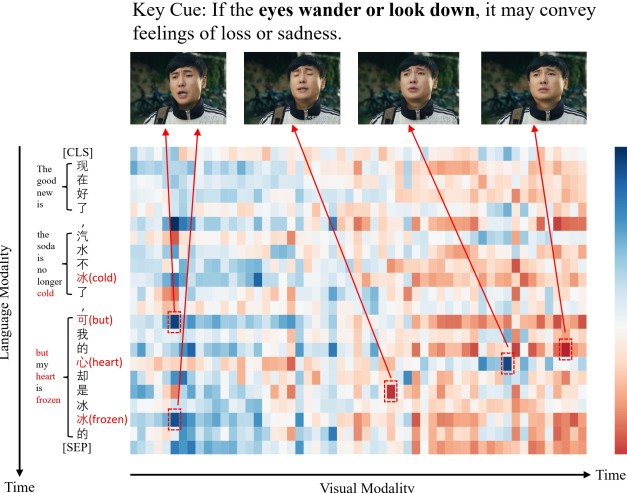

Figure 4: An example of attention difference maps on the SIMS. This difference map is obtained by subtracting the attention map without MLLMs' prompts from the conditional attention map $H_{V \to L}^{Teacher}$. Note: The blue areas indicate regions where the model focuses more when guided by the prompts, while the orange areas indicate regions where the model focuses less under the same prompts.

## B.13   Examples of MLLMs' Prompts

As shown in Figure 5 and Figure 6, we provide more examples of MLLMs' (*i.e.,* GPT-4o-mini and Gemini-2.0-Flash) prompts, both in Chinese and English. For efficiency and cost-effectiveness, we uniformly sample three frames from the video input as the input to the MLLMs, consistent with previous works [2]. Compared with Gemini-2.0-Flash, GPT-4o-mini does not support speech data analysis, its output did not include audio cues.

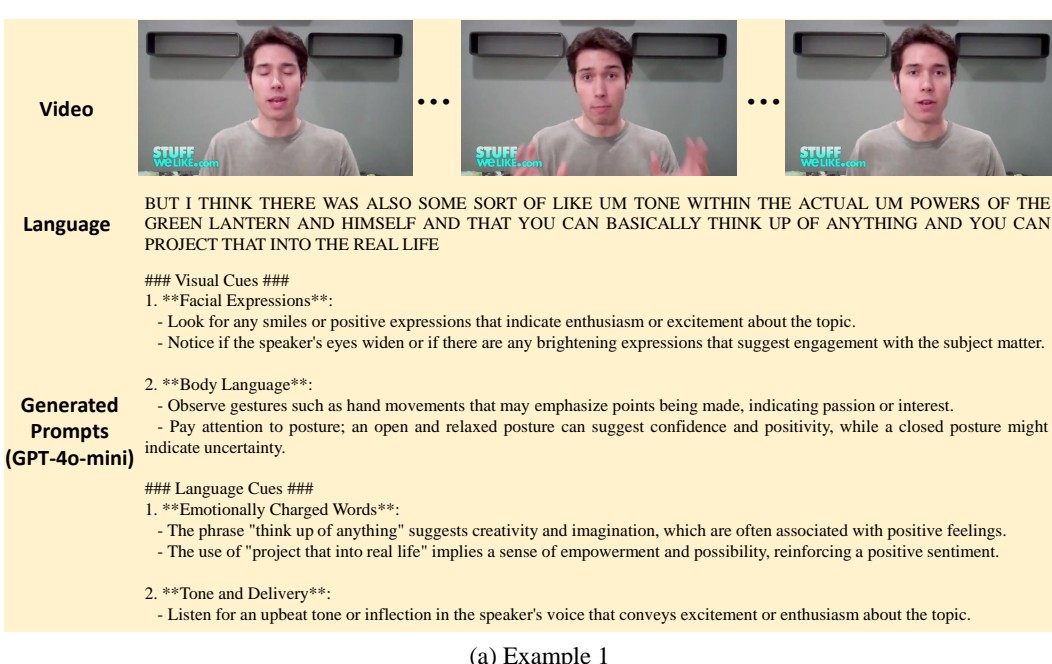

(a) Example 1

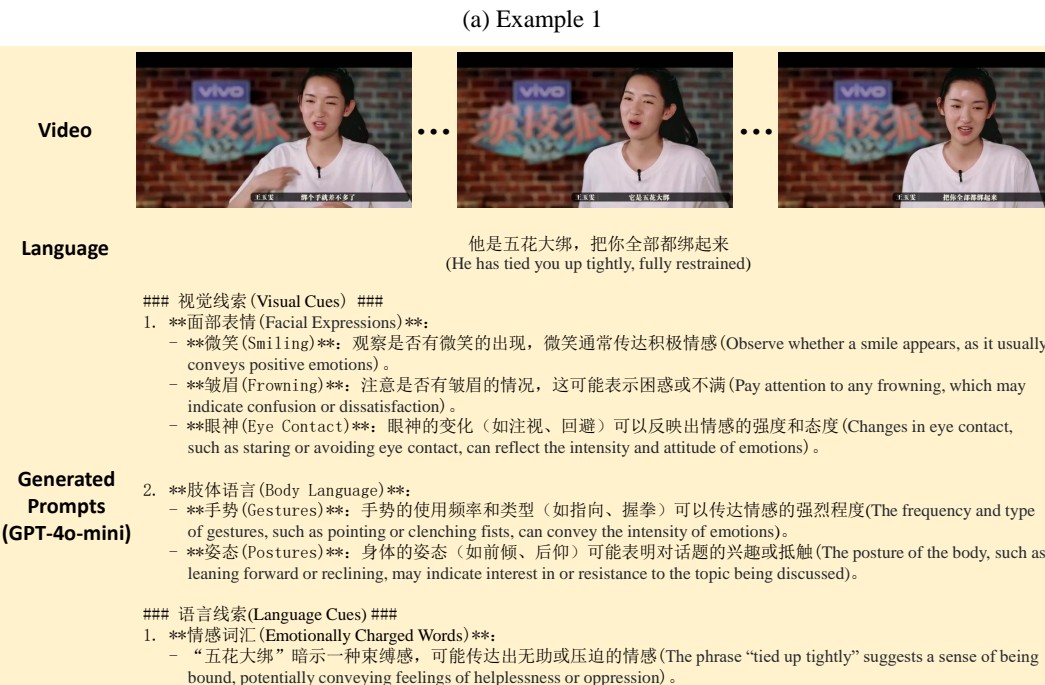

(b) Example 2

Figure 5: Examples of Prompts from GPT-4o-mini.

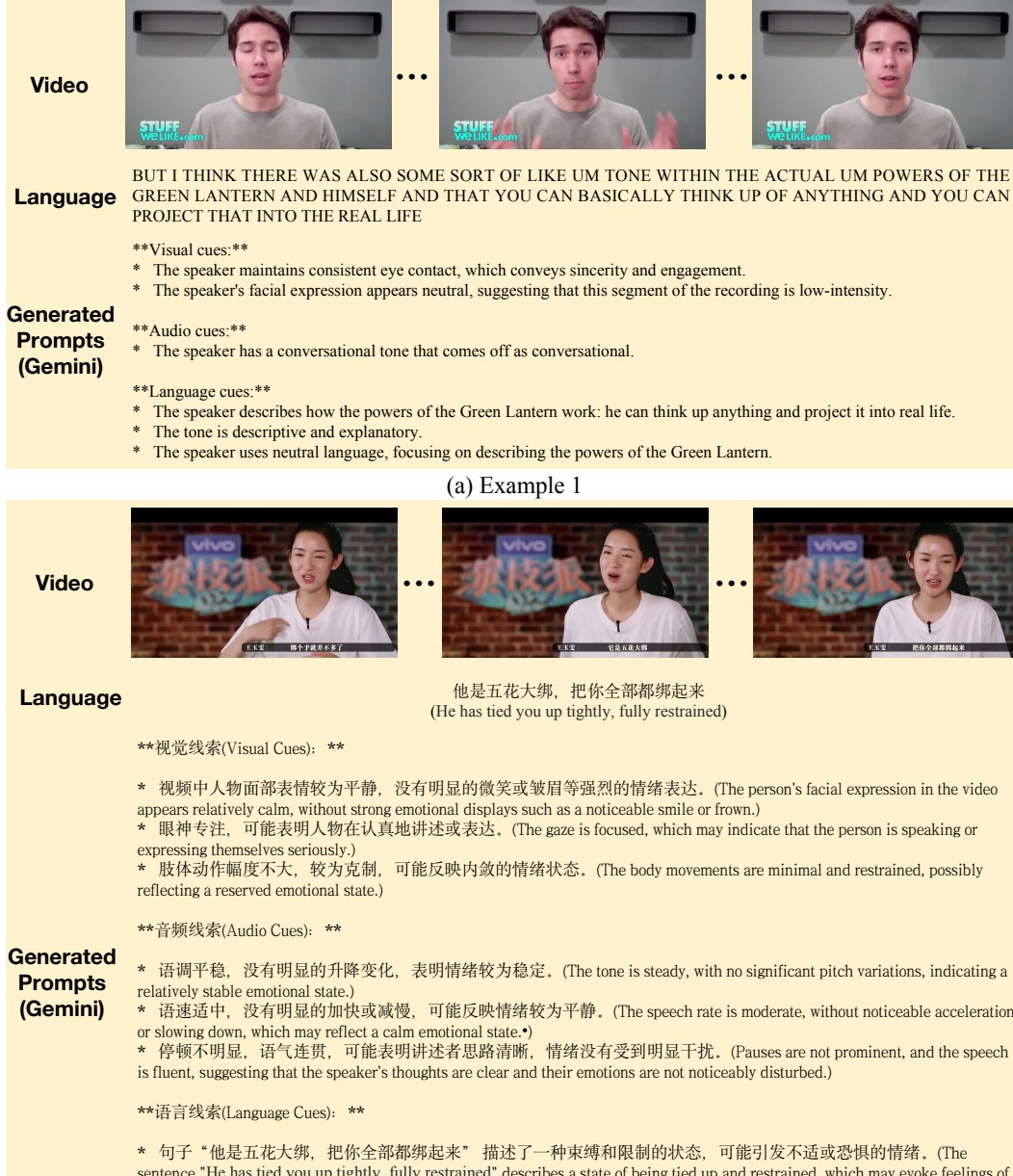

(a) Example 1

(b) Example 2

Figure 6: Examples of Prompts from Gemini-2.0-Flash.

# C  Supplement of Implementation Details

## C.1  Implementation Details and Hyperparameters

We implemented our proposed method using PyTorch 2.1.1 with CUDA 12.1. The experiments were conducted on a PC equipped with an AMD EPYC 7513 processor (2.6GHz) and an NVIDIA Tesla A40 GPU. The key parameters are listed in Table 20.

In the training of the teacher, we perform random mask on the multimodal input to improve the data diversity. The ratio of random masks is between 0 and 70% on the SIMS dataset and between 0 and 50% on the MOSI and MOSEI datasets. Additionally, since GPT-4o-mini does not support speech analysis, we prompted it to infer possible speech cues based on the available language information. The prompt template used for this task is shown in Listing C.1.

Table 20: The parameters used on the SIMS, MOSI and MOSEI datasets

| Parameter | SIMS | MOSI | MOSEI |
|---|---|---|---|
| Common | | | |
| Batch Size | 64 | 64 | 64 |
| Optimizer | AdamW | AdamW | AdamW |
| Epochs | 200 | 200 | 200 |
| Seeds | 1111-1115 | 1111-1115 | 1111-1115 |
| Warm Up | ✓ | ✓ | ✓ |
| Cosine Annealing | ✓ | ✓ | ✓ |
| $d$ | 64 | 64 | 64 |
| $T_{\text{L}}, T_{\text{V}}, T_{\text{A}}, T_{\text{P}}$ | 50, 55, 400, 50 | 50, 500, 375, 50 | 50, 500, 500, 50 |
| The Depth of Language Embedding | 1 | 1 | 1 |
| The Depth of Visual Embedding | 1 | 1 | 1 |
| The Depth of Audio Embedding | 1 | 1 | 1 |
| The Depth of Prompt Embedding | 2 | 2 | 2 |
| MLLMs (GPT-4o-mini) | | | |
| Temperature | 0 | 0 | 0 |
| Version | 2024-07-18 | 2024-07-18 | 2024-07-18 |
| Teacher | | | |
| Initial Learning Rate | 1e-4 | 1e-4 | 2e-4 |
| The Depth of Conditional Alignment | 6 | 6 | 6 |
| The Depth of Multimodal Fusion | 6 | 6 | 6 |
| Student | | | |
| $\alpha, \beta$ | 60.0, 8.0 | 100.0, 4.0 | 100.0, 4.0 |
| Initial Learning Rate | 2e-4 | 1e-4 | 2e-4 |
| The Depth of Conditional Alignment | 2 | 2 | 1 |
| The Depth of Multimodal Fusion | 2 | 2 | 2 |

## C.2  Prompting Template to Generate Prompts for Teacher

Listing C.1 provides the prompting template used to generate prompts for the teacher on the MOSI dataset. Since SIMS is a Chinese dataset, we directly translated this template into Chinese to generate prompts for the teacher on the SIMS dataset. We can see that there is a strong guidance for prediction in the hints given by the MLLMs. Based on these prompts, the teacher is more easily learn the alignment between modalities and in turn transfer this knowledge to the student which does not rely on MLLMs' prompts. More examples can be seen in Appendix B.13.

## Listing C.1: Prompting Template of MLLMs to Generate Prompts for teacher

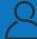 **User**

### Background ###
You are a multimodal sentiment analysis expert. We provide you with a video-text pair and corresponding sentiment label. Please provide detailed hints to help task-specific small models identify sentiment cues.

### Video-text Pair and Corresponding Label ###
**Note: The label is None if the data is from the test set.**
Video: Please refer to the video input.
Text: THAT HUH BEFORE WE GO ON WITH THAT I MUST SAY THAT THIS HAD A SURPRISINGLY DECENT CAST
Label: Positive

### Response Requirements ###
1. Focus on facial expressions in the video, especially smiles, frowns, and eye movements.
2. Focus on the language in the video, especially emotionally charged words and phrases.

### Output Format ###
Your output consists of the following parts:
1. Visual cues: Observations related to facial expressions in the video.
2. Language cues: Important sentiment cues found in the provided text.

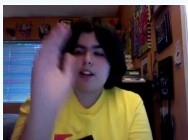 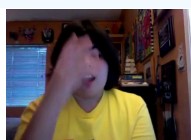 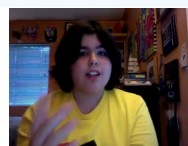

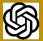 **GPT-4o-mini**

### Visual Cues ###
1. Facial Expressions: - Look for any smiles or positive expressions that indicate enjoyment or enthusiasm. A genuine smile can enhance the positive sentiment.
- Observe the eyes; wide eyes can convey excitement or surprise, while squinting might indicate skepticism.
2. Body Language: - Notice the gestures; open hand movements can suggest openness and positivity. If the individual is using their hands expressively, it may indicate engagement with the topic.
- Pay attention to posture; an upright and relaxed posture can signal confidence and positivity, while slouching might suggest disinterest.

### Language Cues ###
1. Emotionally Charged Words: - The phrase "surprisingly decent cast" carries a positive connotation. The word "surprisingly" suggests an unexpected positive outcome, which can amplify the sentiment.
- The use of "decent" indicates a favorable opinion, suggesting that the speaker is pleasantly surprised by the quality.

## Impact Statement

This paper presents work whose goal is to advance the field of Multimodal Sentiment Analysis and Multimodal Machine Learning. There are many potential societal consequences of our work, none which we feel must be specifically highlighted here.

