# OpenReview forum: "Improving Task-Specific Multimodal Sentiment Analysis with General MLLMs via Prompting"
_NeurIPS.cc/2025/Conference — NeurIPS 2025 poster_

### Official Review · Reviewer_Wv59 · 2025-06-27

**Clarity:** 3
**Significance:** 3
**Originality:** 3
**Rating:** 4
**Confidence:** 4

**Summary:**

This paper focuses on improving multimodal sentiment analysis over language, video & audio inputs and proposes a teacher-student framework that leverages the use of a Multimodal LLM in the teacher model.  The pretrained MLLM is used to generate context-aware prompts focusing on facets in the multimodal inputs that are sentiment-related. The prompts are utilized to conditionally align the different modalities of input (e.g. between video & language), with the aim of capturing aligned sentiment information across the modalities. The outputs of the conditional alignment modules are then fused together and used to predict a sentiment score.
The student model aims to mimic the teacher in terms of the conditional attention map, the outputs of the fusion block, and the regression (sentiment) score.
Comparisons with baselines on 3 datasets shows improved performance across several metrics. Experiments are further conducted to demonstrate the effect of the choice of MLLM in the teacher, as well as the different components of the objective function for the student module.

**Questions:**

1. Are the human annotations referred to in Line 217 directly obtained from the datasets, or was an additional study conducted? If so, could you please provide details on this?
2. Could you please clarify the choice of modeling this as a regression task vs classification? Also, how were the regression scores converted to sentiment labels to compute F1 & Accuracy?
3. Have any experiments been conducted on out of domain datasets?
4. In Table 6, Gemini 2.0 Flash is shown to *underperform* w.r.t GPT 4o mini as the MLLM in the proposed approach. However, in Tables 1 & 2, on the same datasets, by directly using these MLLMs for the task, we find them to be at par for Accuracy & F1, with Gemini  *outperforming* GPT 4o for MAE & Correlation. Any clarifications/thoughts on this occurrence would be helpful to know.

**Ethical Concerns:**

["NO or VERY MINOR ethics concerns only"]

**Final Justification:**

The authors have provided additional details during the rebuttal which will improve the quality of the paper, if included. However, as mentioned during the rebuttal, I would like to remain at my original score at this time.

**Limitations:**

Yes

**Quality:**

3

**Strengths And Weaknesses:**

Strengths:
1. The paper is well-written with clearly explained methodology & experimentation. The in-depth discussion in Section 3 is helpful to understand the details & motivations behind the architecture.
2. Utilizing the MLLM for conditional alignment in the teacher module is a promising and useful strategy to attend to the sentiment cues across various modalities. Leaving it out of the student module enables the usage of the student at inference time to be less resource-intensive which is beneficial.
3. The datasets chosen for experimentation were multilingual (1 Chinese & 2 English) which demonstrates the generalizability of the framework

Weaknesses:
1. The choice of modeling the problem as a regression task over classification over sentiment labels for the teacher-student framework needs some elaboration. Further, it was unclear how sentiment labels were obtained from the regression scores in order to compute Accuracy & F1 scores reported in Tables 1-6.
2. The student module's performance needs improvement if the intention is to only use this module at inference time for computational ease. For all 3 datasets (Tables 1, 2, 3), the teacher module outperforms the student, as would be expected given the use of the MLLM in the teacher. In fact, in Tables 2 & 3, the student's performance is at par or below the baselines compared with.
3. In Table 2, the teacher's performance (presumably using GPT 4o mini as the underlying MLLM given that this is found to be a more effective MLLM in Table 6) is below that of using GPT 4o mini directly for the problem, for all 4 metrics. Similarly, for Table 3, the teacher's performance is at par with the MLLM itself for 3 out of the 4 metrics. This raises questions on the requirement of the added parameters of the teacher module vs using the MLLM directly for the problem.

---

> ### Author Rebuttal · Authors · 2025-07-31
>
> # Response to Reviewer Wv59
>
> **Response to W1**
>
> Thank you for the question. We follow standard practice in MSA and formulate the task as regression because all three benchmark datasets (SIMS, MOSI, MOSEI) provide continuous sentiment scores (SIMS: [-1, 1], MOSI/MOSEI: [-3, 3]). As mentioned in Appendix B, these fine-grained labels support regression-based training. To compute classification metrics (Acc-2, F1), we adopted a threshold-based way which is consistent with previous studies [1, 2]. For example, scores >0 are treated as positive, ≤0 as non-positive (for binary classification). This method is widely used in the MSA studies.
>
> **Response to W2**
>
> Thank you for the insightful comment. **As shown in Table 1 below**, although it is expected that the student may underperform the teacher (which uses MLLM prompts as input), the student achieves competitive results with much fewer parameters and significantly lower inference cost. For example, our student model achieves competitive performance (F1: 81.85±1.41 on SIMS) with only **0.82M parameters**, **8.61 GFLOPs**, and **6.39s** test-time inference. These results demonstrate the potential of our framework that make general knowledge of MLLMs to guide the training of task-specific MSA models, offering new insight into applying general MLLMs to improve MSA. In addition, it is also worth noting that although MMSLF has a small number of parameters, its GFLOPs are relatively higher. This is because we do not compress the input sequences' length as the prior methods [1, 2]. With further optimization (*e.g.,* sequence dimension reduction), the computational cost of MMSLF can be further reduced.
>
> Table 1. Comparison of Efficiency (Evaluate on SIMS dataset). **Bold** results indicates the optimal result. The experiments were conducted on a Computer with an AMD EPYC 7513 CPU and an NVIDIA Tesla A40 GPU.
>
> |||||||
> | --------------- | -------------- | --------------- | ------------------------------------ | ---------------------- | ----------------------- |
> | **Models**      | **Parameters** | **GFLOPs**      | **Inference Time on SIMS Test Set**  | **F1** **on** **SIMS** | **MAE** **on** **SIMS** |
> | GPT-4V          | >7B            | -               | >30min (affected by network quality) | 81.24                  | -                       |
> | GPT-4o-mini     | >7B            | -               | >27min (affected by network quality) | 82.51                  | 0.453                   |
> | Gemini2.0-Flash | >7B            | -               | >22min (affected by network quality) | **84.69**| 0.381                   |
> | TFN             | 35.63M         | **0.10** | **3.46s**                                | 77.83±1.62             | 0.434±1.12              |
> | MISA            | 21.66M         | 7.32     | 12.32s                               | 76.54±1.67             | 0.451±1.83              |
> | Self-MM         | **0.38M**      | 6.66     | 11.40s                               | 77.72±0.68             | 0.418±1.05              |
> | TETFN           | 1.53M          | 6.72     | 26.57s                               | 79.34±0.52             | 0.422±1.30              |
> | ALMT            | 2.60M          | 7.00     | 16.08s                               | 80.17±0.60             | 0.421±0.69              |
> | MMSLF-Teacher   | 2.54M          | 96.16    | 12.31s + 27min                       | 84.06±0.43 | **0.370±0.50**          |
> | MMSLF-Student   | 0.82M          | 8.61     | 6.39s                            | 81.85±1.41             | 0.382±1.39              |
> |||||||
>
> **Response to W3**
>
> Thank you for the observation. The teacher model in our framework serves as a medium to transfer the general knowledge of MLLMs into a smaller student model. While the teacher leverages prompts from a powerful MLLM (e.g., GPT-4o-mini), some performance gap is expected due to information loss during knowledge transfer, which is common in teacher-student frameworks. The purpose of introducing the teacher is not to surpass the MLLM, but to bridge its capabilities into a more efficient, task-specific model. Ultimately, our student model achieves trade-off performance, validating the effectiveness of our idea.
>
> **Response to Q1**
>
> It is directly obtained from the datasets. We will emphasize these points in the revised dataset introduction. Thank you for your question.
>
> **Response to Q2**
>
> Thank you for the question. We model the task as regression because all three benchmark datasets (SIMS, MOSI, MOSEI) provide continuous sentiment annotations (SIMS: [-1, 1], MOSI/MOSEI: [-3, 3]), and regression enables better supervision for fine-grained sentiment intensity. As mentioned in Appendix B, these fine-grained labels support regression-based training. To compute classification metrics (Acc-2, F1), we adopted a threshold-based way which is consistent with previous studies [1, 2]. For example, scores >0 are treated as positive, ≤0 as non-positive (for binary classification). This method is widely used in the MSA studies.
>
> **Response to Q3**
>
> Thank you for the question. In this work, we use preprocessed modality sequence provided by each dataset, which are commonly adopted in prior MSA methods. These feature representations vary across datasets in both dimension and format, making direct cross-dataset evaluation infeasible. We follow this setting to ensure fair comparison with existing baselines.
>
> **Response to Q4**
>
> **As shown in Table 2 below**, we evaluated our method using different prompt templates to assess robustness. We selected GPT-4o-mini and Gemini-2.0-Flash because these larger MLLMs typically exhibit stable outputs across prompt variations, which is essential for reliable knowledge distillation. Our experimental results show minimal performance variance across different prompt formulations within the same MLLM, confirming that advanced large-scale MLLMs generate consistent outputs regardless of reasonable prompt variations. This stability validates our framework's robustness to prompt engineering choices. However, the more significant factor is the selection of an appropriate teacher mLLM with strong domain-specific capabilities. The performance difference between GPT-4o-mini and Gemini-2.0-Flash may stem from their varying abilities to generate accurate and informative prompts for sentiment-centered multimodal tasks, rather than sensitivity to prompt variations. Therefore, we believe this might be due to that Gemini‑2.0- Flash was trained on comparatively less fine‑grained affective data, limiting its ability to craft sentiment‑focused multimodal descriptions. We will include this discussion in the revised manuscript.
>
> Table 2. Comparison of Prompt Sensitivity and Robustness. **Bold** results indicates the optimal result.
> |||||||
> | --------------- | -------------- | --------------- | ------------------------------------ | ---------------------- | ----------------------- |
> | **SIMS**      |                                 |                |                |                |                |
> | Method        | Prompt                          | Acc-2          | F1             | MAE            | Corr           |
> | MMSLF-Teacher | Prompt 1 (Default, GPT-4o-mini) | **83.06±0.95** | **84.06±0.43** | 0.370±0.50     | 0.690±0.80     |
> | MMSLF-Teacher | Prompt 2 (GPT-4o-mini)          | 81.23±5.94     | 83.93±1.16     | **0.346±1.12** | **0.716±1.47** |
> | MMSLF-Teacher | Prompt 3 (GPT-4o-mini)          | 80.04±5.37     | 82.51±0.71     | 0.372±1.20     | 0.687±2.72     |
> | MMSLF-Teacher | Prompt 4 (Gemini-2.0-Flash)     | 81.09±0.23     | 81.09±0.29     | 0.377±0.73     | 0.686±1.53     |
> | MMSLF-Teacher | Prompt 5 (Gemini-2.0-Flash)     | 77.90±7.03     | 82.87±1.21     | 0.355±2.25     | 0.704±3.08     |
> | MMSLF-Teacher | Prompt 6 (Gemini-2.0-Flash)     | 71.42±4.11     | 81.95±0.07     | 0.384±1.47     | 0.658±1.82     |
> | **MOSI**      |                                 |                |                |                |                |
> | Method        | Prompt                          | Acc-2          | F1             | MAE            | Corr           |
> | MMSLF-Student | Prompt 1 (Default, GPT-4o-mini) | **81.40±1.58** | **81.85±1.41** | 0.382±1.39     | 0.662±1.26     |
> | MMSLF-Student | Prompt 2 (GPT-4o-mini)          | **81.40±1.09** | 81.65±0.96     | 0.394±1.06     | 0.667±0.45     |
> | MMSLF-Student | Prompt 3 (GPT-4o-mini)          | 80.74±0.37     | 80.92±0.74     | 0.393±1.80     | 0.664±1.34     |
> | MMSLF-Student | Prompt 4 (Gemini-2.0-Flash)     | 80.00±0.41     | 80.11±0.54     | 0.422±0.96     | 0.627±1.70     |
> | MMSLF-Student | Prompt 5 (Gemini-2.0-Flash)     | 81.18±0.97     | 81.15±0.89     | 0.387±1.92     | 0.670±2.69     |
> | MMSLF-Student | Prompt 6 (Gemini-2.0-Flash)     | 81.05±0.45     | 81.14±0.37     | **0.381±0.96** | **0.676±1.59** |
> |||||||
>
> **Reference**
>
> [1] MISA: modality-invariant and -specific representations for multimodal sentiment analysis. In ACM MM 2020.
>
> [2] Learning language-guided adaptive hyper-modality representation for multimodal sentiment analysis. In EMNLP 2023.

---

> > ### Comment · Reviewer_Wv59 · 2025-08-08
> >
> > I thank the authors for taking the time to provide additional details & answers, which will strengthen the quality of the paper, if included. I am maintaining my original score at this time.

---

> > > ### Author Response · Authors · 2025-08-09
> > >
> > > Dear Reviewer Wv59,
> > >
> > > Thank you very much for your feedback. We will certainly include these additional details and answers in the revised version of the paper, and we believe this will further improve the quality of our paper.
> > >
> > > Best Regards,
> > >
> > > The Authors

---

### Official Review · Reviewer_ctoX · 2025-06-29

**Clarity:** 3
**Significance:** 2
**Originality:** 3
**Rating:** 4
**Confidence:** 5

**Summary:**

The paper introduces MLLM-Guided Multimodal Sentiment Learning Framework (MMSLF), a framework that leverages multi-modal large language models (MLLM) to improve the performance of task-specific multimodal sentiment analysis (MSA) models in a teacher-student setup.  The teacher uses information from a powerful MLLM model (specifically, GPT-4o-mini in this study) to generate context-aware prompts that help the student focus on better alignments between different modalities of the data. Experimental results on 3 benchmark datasets show that the proposed framework leads to competitive task-specific models.

**Questions:**

The authors show that the results obtained with  GPT-4o-mini  are still better than those obtained with Gemini-2.0-Flash. Some insights into why this happens would be useful.

**Ethical Concerns:**

["NO or VERY MINOR ethics concerns only"]

**Final Justification:**

The authors addressed most of my comments and suggestions, and they plan to update the paper accordingly, to the extent the page limit allows. I increased my score.  However, the significance of the contribution is still somewhat weak in my view.

**Limitations:**

The results of  Gemini-2.0-Flash are the best overall. A discussion of the tradeoff between computational resources and performance is lacking.

It is not clear what the assumptions of different baselines might be and what data modalities they work with. That might explain some of the results.

There is no statistical significance analysis.

**Quality:**

3

**Strengths And Weaknesses:**

The proposed framework leverages powerful MLLMs to improve the results of smaller task-specific fine-tuned models. To do that, it makes sense of a teacher-student framework in an interesting way, where the teacher is equipped with an MLLM and generates prompts that help the student align multimodal data and focus on what may be more representative features for the multimodal sentiment analysis task.

The authors run experiments on three datasets and compare their results with the results of a large set of baselines, including MLLMs and also task-specific fine-tuned models.

Experimental results show that the proposed approach has the ability to improve the results of prior fine-tuned models.

The authors motivate the proposed approach by saying that the use of MLLMs for the task of multimodal sentiment analysis may not be justified, given the increase in computational costs required for good performance. However, an MLLM is still used for the teacher. While the student does not directly use output from an MLLM, its performance lags significantly behind that of the teacher and also the performance of other MLLMs (specifically, Gemini-2.0-Flash). It is not clear that the drop in performance is acceptable. The authors do not do any analysis in terms of computational resources used by different models, despite this being part of the motivation for the work.

The authors perform 5 runs for each experiment and report average metrics together with standard deviation. However, there is no statistical significance analysis and it's not clear that the student model is indeed better than most of the other fine-tuned models.

As a side note, there is no discussion of the specific multimodal information that each of the baselines are using. Gemini-2.0-Flash is able to use all the modalities of the data and gives the best results overall, while the MLLM used in the teacher doesn't seem to support audio analysis. The authors show that the results obtained with  GPT-4o-mini  are still better than those obtained with Gemini-2.0-Flash but it's not clear why.

The captions of the tables are not self-explanatory.

---

> ### Author Rebuttal · Authors · 2025-07-31
>
> # Response to Reviewer ctoX
>
> **Response to W1**
>
> Thank you for your valuable comments. **(1) As shown in Table 1**, in the revised manuscript, we will add the following which quantitatively compares advanced MLLMs and our MMSLF in terms of parameter count, GFLOPs, and inference time. Obviously, our student model achieves competitive performance (F1: 81.85±1.41 on SIMS) with only **0.82M parameters**, **8.6 GFLOPs**, and **6.39s** test-time inference. In contrast, Gemini-2.0-Flash and GPT-4o-mini require larger parameters and >22min inference time. These results demonstrate that our method offers a trade-off between performance and computational cost. In addition, it is also worth noting that although MMSLF has a small number of parameters, its GFLOPs are relatively higher. This is because we do not compress the input sequences' length as the prior methods [1, 2]. With further optimization (*e.g.,* sequence dimension reduction), the computational cost of MMSLF can be further reduced.  **(2)** We can see that the Student’s performance is lower than that of the Teacher, GPT‑4o‑mini, and Gemini‑2 Flash because it omits MLLM‑generated prompts and inevitably loses some information during distillation. However, this design itself confirms the feasibility of using general MLLMs to improve a much smaller model’s performance. **(3)** Although there remains a gap between the Student and GPT‑4o‑mini or Gemini‑2.0-Flash, the student still surpasses other large models such as GPT‑4V. In summary, we believe our motivation is sound and the corresponding exploration meaningful, and we will further clarify this in the revised manuscript. In addition, due to the limitation on the number of rebuttal words. Overall, these results demonstrate that our method offers a trade-off between performance and computational cost.
>
> Table 1. Comparison of Efficiency (Evaluate on SIMS dataset). **Bold** results indicates the optimal result. The experiments were conducted on a Computer with an AMD EPYC 7513 CPU and an NVIDIA Tesla A40 GPU.
>
> |||||||
> |-|-|-|-|-|-|
> | Models |Parameters|GFLOPs| Inference Time on SIMS Test Set |F1 on SIMS | MAE on SIMS|
> | GPT-4V          | >7B            | -               | >30min (affected by network quality) | 81.24                  | -                       |
> | GPT-4o-mini     | >7B            | -               | >27min (affected by network quality) | 82.51                  | 0.453                   |
> | Gemini2.0-Flash | >7B            | -               | >22min (affected by network quality) |**84.69**| 0.381                   |
> | TFN             | 35.63M         | **0.106288368** | **3.46s**                                | 77.83±1.62             | 0.434±1.12              |
> | MISA            | 21.66M         | 7.327169664     | 12.32s                               | 76.54±1.67             | 0.451±1.83              |
> | Self-MM         | **0.38M**      | 6.657025504     | 11.40s                               | 77.72±0.68             | 0.418±1.05              |
> | TETFN           | 1.53M          | 6.720118224     | 26.57s                               | 79.34±0.52             | 0.422±1.30              |
> | ALMT            | 2.60M          | 7.004771456     | 16.08s                               | 80.17±0.60             | 0.421±0.69              |
> | MMSLF-Teacher   | 2.54M          | 96.159077376    | 12.31s + 27min |84.06±0.43| **0.370±0.50**          |
> | MMSLF-Student   | 0.82M          | 8.610916288     | 6.39s| 81.85±1.41             | 0.382±1.39              |
> |||||||
>
> **Response to W2**
>
> Thank you for pointing this out. **As shown in Table 2 below,** we have added an experiment: on the SIMS dataset we conducted two‑tailed *t*-tests between the Student model and the best task‑specific baseline. The resulting *p*-values are MAE = 0.0029, Corr = 0.00022, Acc‑2 = 0.095, and F1 = 0.066. Under the conventional 0.05 threshold, the Student’s improvements on MAE and Corr are statistically significant, while the gains on ACC‑2 and F1 show the same positive trend. Because of time and rebuttal‑length constraints, we will include a more detailed comparison on all datasets in the revised manuscript.
>
> Table 2. Two‑tailed *t*-tests between the Student model and the ALMT on SIMS dataset.
>
> ||||||
> |-|-|-|-|-|
> | Models           | p-value (MAE) | p-value (Corr) | p-value(Acc-2) | p-value (F1) |
> | MMSLF-Student & ALMT | 0.0029                        | 0.00022                | 0.095                   | 0.066                        |
> ||||||
>
> **Response to W3**
>
> Thank you for raising this point. (1) In Section 4.8, we mentioned that "the prompts may contain more misleading information, which may result in poorer performance when using prompts generated by Gemini-2.0-Flash for model training as discussed in Section 4.7. (2) We have provided one case (you can also refer to Case 1 in Appendix Figure 6), and the descriptions generated by Gemini sometimes contain errors. (3) **As shown in Table 3 below**, we use different templates to generate prompts and then use them to train the model. The results show that minimal performance variance across different prompt formulations within the same MLLM, confirming that advanced large-scale MLLMs generate consistent outputs regardless of reasonable prompt variations. Therefore, we believe this might be due to the fact that Gemini‑2.0- Flash was trained on comparatively less fine‑grained affective data, limiting its ability to craft sentiment‑focused multimodal descriptions. We will include this discussion in the revised manuscript.
>
> > Case 1 in Appendix Figure 6:
> >
> > Sentiment Label: Positive
> >
> > Audio Prompts of Gemini-2.0-Flash: The speaker has a conversational tone that comes off as conversational.
>
> Table 3. Comparison of Prompt Sensitivity and Robustness. **Bold** results indicates the optimal result.
>
> |||||||
> |-|-|-|-|-|-|
> | **SIMS**      |                                 |                |                |                |                |
> | Method        | Prompt                          | Acc-2          | F1             | MAE            | Corr           |
> | MMSLF-Teacher | Prompt 1 (Default, GPT-4o-mini) | **83.06±0.95** | **84.06±0.43** | 0.370±0.50     | 0.690±0.80     |
> | MMSLF-Teacher | Prompt 2 (GPT-4o-mini)          | 81.23±5.94     | 83.93±1.16     | **0.346±1.12** | **0.716±1.47** |
> | MMSLF-Teacher | Prompt 3 (GPT-4o-mini)          | 80.04±5.37     | 82.51±0.71     | 0.372±1.20     | 0.687±2.72     |
> | MMSLF-Teacher | Prompt 4 (Gemini-2.0-Flash)     | 81.09±0.23     | 81.09±0.29     | 0.377±0.73     | 0.686±1.53     |
> | MMSLF-Teacher | Prompt 5 (Gemini-2.0-Flash)     | 77.90±7.03     | 82.87±1.21     | 0.355±2.25     | 0.704±3.08     |
> | MMSLF-Teacher | Prompt 6 (Gemini-2.0-Flash)     | 71.42±4.11     | 81.95±0.07     | 0.384±1.47     | 0.658±1.82     |
> | **MOSI**      |                                 |                |                |                |                |
> | Method        | Prompt                          | Acc-2          | F1             | MAE            | Corr           |
> | MMSLF-Student | Prompt 1 (Default, GPT-4o-mini) | **81.40±1.58** | **81.85±1.41** | 0.382±1.39     | 0.662±1.26     |
> | MMSLF-Student | Prompt 2 (GPT-4o-mini)          | **81.40±1.09** | 81.65±0.96     | 0.394±1.06     | 0.667±0.45     |
> | MMSLF-Student | Prompt 3 (GPT-4o-mini)          | 80.74±0.37     | 80.92±0.74     | 0.393±1.80     | 0.664±1.34     |
> | MMSLF-Student | Prompt 4 (Gemini-2.0-Flash)     | 80.00±0.41     | 80.11±0.54     | 0.422±0.96     | 0.627±1.70     |
> | MMSLF-Student | Prompt 5 (Gemini-2.0-Flash)     | 81.18±0.97     | 81.15±0.89     | 0.387±1.92     | 0.670±2.69     |
> | MMSLF-Student | Prompt 6 (Gemini-2.0-Flash)     | 81.05±0.45     | 81.14±0.37     | **0.381±0.96** | **0.676±1.59** |
> |||||||
>
> **Response to W4**
>
> Thank you for pointing this out. In the revised manuscript we will improve the table caption. For example, stating the brief information and explanations of any symbols or abbreviations.
>
> **Response to Q1**
>
> Please see the **Response to W3**. We will include a more detailed comparison in the revised manuscript.
>
> **Response to L1**
>
> Thank you for this question. **As mentioned in Response to W1 and the corresponding table**, although Gemini‑2 .0-Flash delivers the advanced performance, it relies on an external 7 B‑parameter API whose end‑to‑end inference exceeds 22 min and is bounded by network latency. By contrast, with the guidance of the teacher during the training stage, our MMSLF‑Student achieve competitive performance with fewer parameters, fewer GFLOPs, and fewer inference time. Similar phenomena can also be observed in GPT-4o-mini. These results illustrate that our framework offers a new insight into real-world applications of MSA.
>
> **Response to L2**
>
> Thanks for your suggestions! We will specifically mention this in the revised paper. Actually, all task‑specific models, as well as Gemini‑2.0‑Flash and Video‑LLaMA, take vision, audio, and language as inputs. GPT‑4o‑mini and GPT‑4V take vision and text as inputs because they do not support audio analysis.
>
> **Response to L3**
>
> **As shown in Response to W2**, we have added the significance test to compare with the suboptimal model ALMT. We will include a more detailed comparison in the revised manuscript.
>
> **Reference**
>
> [1] MISA: modality-invariant and -specific representations for multimodal sentiment analysis. In ACM MM 2020.
>
> [2] Learning language-guided adaptive hyper-modality representation for multimodal sentiment analysis. In EMNLP 2023.

---

> > ### Comment · Reviewer_ctoX · 2025-08-06
> >
> > Thank you for responding to all my questions and comments. I believe the additions you mentioned will help improve the paper and I updated my score accordingly.

---

> > > ### Author Response · Authors · 2025-08-07
> > >
> > > Dear Reviewer ctoX,
> > >
> > > We sincerely thank you for the encouraging follow-up and for increasing the score! We are greatly encouraged by the opportunity to address your concerns. The relevant results and discussion will be included in the revised paper.
> > >
> > > Best Regards,
> > >
> > > The Authors

---

### Official Review · Reviewer_EAd9 · 2025-07-03

**Clarity:** 2
**Significance:** 3
**Originality:** 3
**Rating:** 4
**Confidence:** 5

**Summary:**

The authors introduces MMSLF (mLLM-guided sentiment analysis) where they feed mLLM output guidance as additional information into a (pretty much) standard sentiment analysis model to improve performance. Further, this model is used as a teacher to distill information into a sentiment analysis model that does not require mLLM input, thus significantly saving computational resources at inference. Performance for the mLLM+sentiment analysis model is close to SOTA for SIMS (especially), MOSI, and MOSEI datasets, while the performance for the teacher is still pretty good. The authors run a comparison with most of the popular MM fusion models/algorithms and perform a comprehensive ablation study.

**Questions:**

1. Could you use this on top a very strong baseline model - like the one mentioned above, how dependent is your mLLM prompt mixing approach on your AVT fusion architecture
2. Prompt Sensitivity and Robustness: Given the significant performance difference between GPT-4o-mini and Gemini-2.0-Flash prompts, how sensitive is the method to prompt variations?
3. Computational Cost Analysis: While you mention cost-effectiveness, can you provide concrete numbers comparing the total computational cost of your approach - also vs other SOTA methods like the ones i mentioned above
4. How relevant is your approach for other multimodal tasks?

**Ethical Concerns:**

["NO or VERY MINOR ethics concerns only"]

**Final Justification:**

The additional experiments provided by the authors help resolve many of my concerns.

**Limitations:**

yes

**Quality:**

3

**Strengths And Weaknesses:**

Strengths:
1. Novel and Practical Approach: the idea of leveraging  mLLM knowledge via text output and then distilling this information in a teacher -student framework is novel.
2. Comprehensive Evaluation: The paper includes thorough experiments across three datasets, multiple baselines (both traditional and mLLM-based), and extensive ablation studies examining each component's contribution.
3. Technical Correctness: The authors do all the reasonable things without being especially novel in their MM architecture. The distillation is the most interesting part.
4. Clear Motivation and Results: The cost-effectiveness argument is compelling and practical - the contribution of mLLM output is clear via the ablation study. Overall, persuasive results

Weaknesses:
1. The abstract and intro do little service to the paper. The main idea is not clearly exposed, i.e., explain what is the information that you are adding and in what format, explain the novelty of the paper in the distillation etc. Conclusions are a bit better. Still thorough rewrite is needed.
2. You should at least explain where you sit in terms of mLLM SOTA, e.g., see this recent paper https://arxiv.org/pdf/2504.11082  -  in general your results are good but you are giving the impression that you are SOTA which is not true.
3. Your literature review is ok and the methods you describe are solid - but i would like to see more focus on papers that introduce new ideas rather than having a ton of top performers in your studies, it would be nice to mabe also review ideas from https://lab-msp.com/MSP-Podcast_Competition/IS2025/
4. The fusion and in general the architecture is of little novelty; however, the teacher-model part is interesting
5. mLLM dependency is a concern - how you engineer the mLLM output is super-important and it is hard to tune as one can see from results from different mLLLMs
6. Without the mLLM prompts results are a low - it is unclear to me if this a  weakness of the architecture or lack of tuning - please explain
7. The method is not modular enough to be applied on top of any existing sentiment analysis model at its current form

---

> ### Author Rebuttal · Authors · 2025-07-31
>
> # Response to Reviewer EAd9
>
> **Response to W1**
>
> We sincerely thank your feedback. Since the submission was only 9 pages long, some details may not be included. In the revised paper, we will: **(1)** Explain the information and format contained in the prompts. **(2)** Emphasize the innovation of the framework. **(3)** Clarify our ideas more clearly, which refer to using general knowledge from large-scale language models to guide the training of MSA models for specific tasks. Please let us know if you have any more specific suggestions. Thank you very much!
>
> **Response to W2**
>
> Thank you for bringing this important recent work to our attention. **(1)** We missed this concurrent work (arXiv:2504.11082), which was released just one month before the NeurIPS submission deadline. We will add comparisons in our revision. **(2) As shown in Table 1 in Response to Reviewer RVt6**, we include the methods for comparison. We compare methods without open-sourced code (i.e., TCAN, DeepMLF) using reported best-run results. For those with open-sourced code (i.e., MAG-BERT, DMD, DLF, MMML), since the limited time of rebuttal, we will reproduce five-run averages under the same setting on all datasets in the revision. Obviously, the results show our MMSLF achieves competitive/better performance in many metrics compared with task-specific models. For example, both teacher and student outperform the work DLF on MOSI, while DLF performs better on MOSEI. Compared to DeepMLF (MLLM), the student underperforms on several metrics. We believe this is due to the fact that part of the knowledge transfer from the MLLM and the teacher may be lost due to the student’s limited capacity. Despite this, our method achieves competitive performance, demonstrating a favorable trade-off. We will discuss these works in the revised version.
>
> **Response to W3**
>
> We sincerely thank you for your sharing and suggestions! In our revision, we will: **(1)** Expand the literature review to discuss novel conceptual contributions and emerging ideas in the field. **(2)** Include deeper discussion of the new insights we discovered, incorporating the key points raised in our responses to all three reviewers.
>
> **Response to W4**
>
> We sincerely thank you for your recognition of our teacher model design. Regarding the student architecture, we would like to clarify: **(1)** While the pipeline is relatively straightforward, this design choice was made based on empirical findings that this structure can achieve distillation efficiency for our specific task. The simplicity of the architecture provides a clearer demonstration of our method's effectiveness by minimizing confounding factors from complex architectural innovations. **(2) As shown in Table 1 below**, we conducted preliminary experiments attempting to directly integrate MLLM prompts into the existing ALMT model. Obviously, ALMT-Teacher achieved the best performance, but ALMT-Student performed worse than MMSLF-Student. We believe that architectural simplicity is not merely a limitation but rather an informed design choice that balances effectiveness with training stability. **(3) As mentioned in lines 56-57 in the paper**, we believe the contribution lies in our whole framework that explores using the general knowledge of MLLMs to guide the training of task-specific MSA models, offering a new insight into applying general MLLMs to improve MSA.
>
> Table 1. Generality of the proposed framework. **Bold** results indicates the optimal result.
>
> ||||||
> |-|-|-|-|-|
> |**SIMS**|||||
> |Method| Acc-2|F1|MAE|Corr|
> |ALMT-Teacher|**84.20±0.57**|**84.45±0.81**|**0.363±0.76**|**0.711±1.50**|
> |ALMT-Student|79.87±1.81|80.58±1.05|0.418±2.15|0.587±3.97|
> |MMSLF-Teacher|83.06±0.95|84.06±0.43|0.370±0.50|0.690±0.80|
> |MMSLF-Student|81.40±1.58|81.85±1.41|0.382±1.39|0.662±1.26|
> |**MOSI**|||||
> |Method| Acc-2|F1|MAE|Corr|
> |ALMT-Teacher|**86.56±0.68**/**88.02±0.67**|**86.63±0.69**/**88.06±0.68**|**0.677±0.57**|**0.834±0.46**|
> |ALMT-Student|83.26±0.41/85.43±0.14|83.38±0.31/85.52±0.15|0.720±0.54|0.784±0.28|
> |MMSLF-Teacher|85.05±0.66/86.61±0.69|85.15±0.66/86.69±0.69|0.734±1.46|0.797±0.60|
> |MMSLF-Student|83.62±0.91/85.37±1.00|83.68±0.96/85.50±0.96|0.746±1.63|0.775±1.10|
> ||||||
>
> **Response to W5**
>
> Thank you for raising this concern. The choice of MLLM significantly impacts performance through the quality of generated guidance signals. **As discussed in Section 4.8**, our analysis reveals that Gemini-2.0-Flash produces less accurate and informative prompts compared to GPT-4o-mini, reflecting its weaker capabilities in sentiment-aware multimodal understanding. However, **(1)** We provide our detailed prompt templates in our appendix. **(2)** As general MLLMs continue to advance rapidly, we believe that this dependency will become less problematic. We will discuss this in our revision.
>
> **Response to W6**
>
> This phenomenon is caused by the architectural constraints. **(1)** To effectively learn from MLLM prompts, our teacher model requires sufficient capacity. **As shown in Table 8 in the Appendix,** we set it to 12 layers. When mLLM prompts are removed, this relatively large teacher model suffers from severe overfitting on the limited MSA training data, leading to poor generalization on test sets. **(2)** While the student model is parameter-efficient, it lacks the representational capacity to achieve competitive performance without teacher guidance. Therefore, our current design represents a carefully calibrated trade-off between performance and parameters. We will clarify this in the revised version.
>
> **Response to W7**
>
> While our current implementation has constraints that limit direct plug-and-play application, we believe this represents an important first step toward a broader research direction. **(1)** Our idea can be adapted to existing MSA models. For example, we tested this on a strong baseline ALMT. Specifically, we concatenate prompts with datasets' text inputs followed by dimensionality reduction. **As shown in Table 1 above**, although the final performance was also good, we observed slower convergence and suboptimal performance compared to our proposed MMSLF-Student model. **(2)** Despite these constraints of direct plug-and-play application, our work demonstrates the potential of utilizing MLLMs to improve task-specific model learning. We aim to validate this research direction and offer new insight into applying general MLLMs to improve MSA. We will discuss this in the revised paper.
>
> **Response to Q1**
>
> We attempted to apply our framework to the recent work (arXiv:2504.11082). However, no open-source implementation was available for this method. Therefore, we present experimental results using ALMT, a strong baseline in the MSA domain. More detailed discussion can be found in **our response to W4, W7 and Table 1 above.**
>
> **Response to Q2**
>
> **As shown in Table 2 below**, we evaluated our method using different prompt templates to assess robustness. We selected GPT-4o-mini and Gemini-2.0-Flash because these larger MLLMs typically exhibit stable outputs across prompt variations, which is essential for reliable knowledge distillation. Our experimental results show minimal performance variance across different prompt formulations within the same MLLM, confirming that advanced large-scale MLLMs generate consistent outputs regardless of reasonable prompt variations. This stability validates our framework's robustness to prompt engineering choices. However, the more significant factor is the selection of an appropriate teacher mLLM with strong domain-specific capabilities. The performance difference between GPT-4o-mini and Gemini-2.0-Flash may stem from their varying abilities to generate accurate and informative prompts for sentiment-centered multimodal tasks, rather than sensitivity to prompt variations.
>
> Table 2. Comparison of Prompt Sensitivity and Robustness. **Bold** results indicates the optimal result.
> |||||||
> |-|-|-|-|-|-|
> |**SIMS**|||||
> |Method|Prompt |Acc-2|F1|MAE|Corr|
> |MMSLF-Teacher|Prompt 1 (Default, GPT-4o-mini)|**83.06±0.95**|**84.06±0.43**|0.370±0.50|0.690±0.80|
> |MMSLF-Teacher|Prompt 2 (GPT-4o-mini)|81.23±5.94|83.93±1.16|**0.346±1.12**|**0.716±1.47**|
> |MMSLF-Teacher|Prompt 3 (GPT-4o-mini)|80.04±5.37|82.51±0.71|0.372±1.20|0.687±2.72|
> |MMSLF-Teacher|Prompt 4 (Gemini-2.0-Flash)|81.09±0.23|81.09±0.29|0.377±0.73|0.686±1.53|
> |MMSLF-Teacher|Prompt 5 (Gemini-2.0-Flash)|77.90±7.03|82.87±1.21|0.355±2.25|0.704±3.08|
> |MMSLF-Teacher|Prompt 6 (Gemini-2.0-Flash)|71.42±4.11|81.95±0.07|0.384±1.47|0.658±1.82|
> |**MOSI**|||||
> |Method|Prompt |Acc-2|F1|MAE|Corr|
> |MMSLF-Student|Prompt 1 (Default, GPT-4o-mini) |**81.40±1.58**|**81.85±1.41**|0.382±1.39|0.662±1.26|
> |MMSLF-Student|Prompt 2 (GPT-4o-mini)|**81.40±1.09**|81.65±0.96|0.394±1.06|0.667±0.45|
> |MMSLF-Student|Prompt 3 (GPT-4o-mini)|80.74±0.37|80.92±0.74|0.393±1.80|0.664±1.34|
> |MMSLF-Student|Prompt 4 (Gemini-2.0-Flash)|80.00±0.41|80.11±0.54|0.422±0.96|0.627±1.70|
> |MMSLF-Student|Prompt 5 (Gemini-2.0-Flash)|81.18±0.97|81.15±0.89|0.387±1.92|0.670±2.69|
> |MMSLF-Student|Prompt 6 (Gemini-2.0-Flash)|81.05±0.45|81.14±0.37|**0.381±0.96**|**0.676±1.59**|
> |||||||
>
> **Response to Q3**
>
> **As mentioned in Section Introduction**, cost-effectiveness is mainly compared with MLLMs. Because directly applying MLLMs to MSA is costly. To make a more comprehensive comparison, we have also provided the parameter comparison with the task-specific model. As for the work (arXiv:2504.11082) you mentioned, since there is no open-source code available, we are unable to provide a comparison. Due to the limitation on the number of rebuttal words, the detailed results and discussion can be found in **Response to Cons2 and Table 1 in Response to Reviewer RVt6**.
>
> **Response to Q4**
>
> We have extended our framework to Emotion Recognition and Sentiment Analysis in subsequent work, with preliminary results demonstrating that appropriately selected MLLMs can effectively guide task-specific model learning across different domains.

---

> > ### Comment · Reviewer_EAd9 · 2025-08-06
> > **Reply to Authors**
> >
> > I appreciate the authors very detailed and to-the-point replies. The additional experiments, efficiency analysis, and ALMT generalizability tests effectively address most of my technical concerns. The results help clarify the work and improve on the paper's contribution. While some presentation clarity issues and method limitations remain (e.g., prompt tuning), the substantive technical contributions are now better supported. Based on these additions and clarifications, I will raise my score by 1.

---

> > > ### Author Response · Authors · 2025-08-07
> > >
> > > Dear Reviewer EAd9,
> > >
> > > Thank you for your thoughtful review and we sincerely appreciate your positive feedback and the updated rating! We will include the relevant results and discussion in the revised paper.
> > >
> > > Best Regards,
> > >
> > > The Authors

---

### Official Review · Reviewer_RVt6 · 2025-07-03

**Clarity:** 4
**Significance:** 4
**Originality:** 2
**Rating:** 4
**Confidence:** 5

**Summary:**

This paper proposes a distillation-based method to introduce multimodal LLM (MLLM) to the multimodal sentiment analysis (MSA) task. The teacher model is trained with the context prompt from MLLM, called conditional alignment in this paper. Finally, experiments are conducted based on the three popular MSA datasets.

**Questions:**

1. For the MSA task, the THUAIR repo. is a good reference, but it is not all. It is necessary to discuss more papers published after 2023 so that this paper claims the right thing.

2. How to obtain all the prompts when training the teacher model? Is it obtained once, or sample and generate the prompt for each data point?

**Ethical Concerns:**

["NO or VERY MINOR ethics concerns only"]

**Final Justification:**

Most of my concerns have been addressed in the rebuttal. However, I still believe the paper somewhat overstates its contributions. In particular, the comparison with MMML is not reported. If the paper intends to claim a state-of-the-art (SOTA) result, it should demonstrate superior performance compared to MMML; without such evidence, the SOTA claim is not accurate.

That said, achieving SOTA performance is not a prerequisite for publication. In my view, the most important aspects are the uniqueness of the insights and the strength of the experimental support. Therefore, I recommend that the authors revise or qualify the SOTA claim if the results do not surpass those of MMML.

Since the reported results are not better than those of MMML, even when considering the best run, I still regard the paper as borderline acceptable at most, despite the fact that most of my earlier concerns have been addressed.

**Limitations:**

Yes

**Quality:**

3

**Strengths And Weaknesses:**

Pros:

1. The motivation and insight of this paper are interesting. MLLM is a popular technique; how to introduce MLLM to the MSA task is a good motivation.

2. The experiments are conducted based on popular datasets. CMU-MOSEI, CMU-MOSI, and SIMS are the three popular MSA datasets.

3. The paper is well-organized and well-written.

Cons:

1. This paper misses many related works. MAG-BERT[1], DMD[2], DLF[3], TCAN[4], MMML[5]. The performance of this paper is not SOTA. Although the SOTA result is not necessary for publication at the top-tier conference, it is not appropriate to claim that this paper achieves the SOTA performance. All the following papers should be included when discussing MSA tasks. If not, please show reasonable reasons for that. The MMML used another encoder for text; for fair comparison, it is better to use the same encoder for text.

[1]  "Integrating multimodal information in large pretrained transformers." ACL 2020.

[2] "Decoupled multimodal distilling for emotion recognition." CVPR 2023.

[3]  "DLF: Disentangled-language-focused multimodal sentiment analysis." AAAI 2025.

[4] "TCAN: Text-oriented cross attention network for multimodal sentiment analysis." Arxiv 2025

[5] "Multimodal multi-loss fusion network for sentiment analysis." arXiv 2023

2. For teacher-student models, or distillation-based models, efficient metrics are generally needed. Such as the model size, inference time or FLOPs.

3. There is no visualization analysis to show the unique contribution of this paper.

---

> ### Author Rebuttal · Authors · 2025-07-31
>
> # Response to Reviewer RVt6
>
> **Response to Cons 1**
>
> Thank you for the suggestion. **It should be noticed that five runs results** **are** **usually lower than searched best-run results.** We have realized that including more recent methods can strengthen our claims. **As shown in the Table 1 below**, we compare methods without open-sourced code (i.e., TCAN, DeepMLF) using reported best-run results. For those with open-sourced code (*i.e.,* MAG-BERT, DMD, DLF), since the limited time of rebuttal, we will reproduce five-run averages of these methods under the same setting on all datasets in the revision. Obviously, the results show our MMSLF achieves competitive/better performance in many metrics when compared with task-specific models on MOSI, MOSEI and SIMS datasets. For example, both teacher and student outperform the work DLF on MOSI, while DLF performs better on MOSEI. Compared to DeepMLF (task-specific MLLM), the student underperforms on several metrics. We believe this is primarily due to the fact that part of the knowledge transfer from the MLLM and the teacher may be lost due to the student’s limited capacity. Despite this, our method achieves competitive performance, demonstrating a favorable trade-off. We will also discuss the recent/contemporaneous works (*i.e.,* DLF and TCAN) in the revised version in section relevant work.
>
> In addition, regarding MMML, since it employs stronger encoders for text and audio feature extraction, we are working toward a fair comparison under a consistent setting. Specifically, we adopt RoBERTa for text encoding and HuBERT for audio encoding, and we will report the average performance over three runs. **These experiments are currently in progress, and we will update the results on OpenReview once they are available.**
>
> Table 1. Performance comparison. **Bold** results indicates the optimal result.
> | | | | | |
> | ----------------- | -------------- | -------------- | -------------- | -------------- |
> |**SIMS**|||||
> |Method|Acc-2|F1|MAE|Corr|
> | DeepMLF (2025-04, Task-specific MLLM)| 82.75| 83.15| **0.353**|**0.729**|
> | MMSLF-Teacher (Best Run)|**84.25** | **84.16**|0.369|0.693|
> | MMSLF-Student (Best Run)|83.37| 83.51| 0.367|0.681|
> | MMSLF-Teacher (Five Runs)|83.06±0.95| 84.06±0.43|0.370±0.50|0.690±0.80|
> | MMSLF-Student (Five Runs|81.40±1.58| 81.85±1.41|0.382±1.39|0.662±1.26|
> |**MOSI**|||||
> |Method|Acc-2|F1|MAE|Corr|
> | DeepMLF (2025-04, Task-specific MLLM)| -/85.60 | -/85.58| **0.692**  | 0.811|
> | MAG-BERT (2020, Task-specific) | -/86.10| -/86.00| 0.712 | 0.796|
> | DMD (2023, Task-specific) | -/83.23| -/83.29| 0.752| - |
> | DLF (2025-02, Task-specific)   | -/85.06| -/85.04 | 0.731| 0.781|
> | TCAN (2025-04, Task-specific)  | -/86.28 | -/86.15|0.714| 0.797|
> | MMSLF-Teacher (Best Run)| **86.15/87.80** | **86.17/87.89**| 0.700| **0.812**  |
> | MMSLF-Student (Best Run)| 84.55/87.04| 84.73/87.14| 0.701| 0.795|
> | MMSLF-Teacher (Five Runs)| 85.05±0.66/86.61±0.69 | 85.15±0.66/86.69±0.69 | 0.734±1.46 | 0.797±0.60 |
> | MMSLF-Student (Five Runs)| 83.62±0.91/85.37±1.00 | 83.68±0.96/85.50±0.96 | 0.746±1.63 | 0.775±1.10 |
> |**MOSEI**|||||
> |Method|Acc-2|F1|MAE|Corr|
> | DeepMLF (2025-04, Task-specific MLLM) | -/**87.15** | -/87.10| **0.499**  | **0.804**  |
> | MAG-BERT(2020, Task-specific) |-| -| -| -|
> | DMD (2023, Task-specific)| -/84.62| -/84.62| 0.543| - |
> | DLF (2025-02, Task-specific)| -/85.42| -/85.27| 0.536| 0.764 |
> | TCAN (2025-04, Task-specific)| -/86.27| -/86.17| 0.532| 0.774 |
> | MMSLF-Teacher (Best Run)| **85.47/87.15** | **85.70/87.26**| 0.530      | 0.784      |
> | MMSLF-Student (Best Run)| 84.48/85.36  | 84.85/85.43 | 0.542      | 0.755      |
> | MMSLF-Teacher (Five Runs)| 85.08±0.36/86.62±0.75 | 85.55±0.24/86.71±0.71 | 0.539±1.06 | 0.773±1.51 |
> | MMSLF-Student (Five Runs)| 83.96±0.38/84.67±0.27 | 84.20±0.48/84.74±0.28 | 0.548±0.41 | 0.747±0.51 |
> | | | | |
>
> **Response to Cons 2**
>
> Thank you for the suggestion. We provide a detailed comparison in **Table 2**. Notably, our student model achieves competitive performance (F1: 81.85±1.41 on SIMS) with only **0.82M parameters**, **8.61 GFLOPs**, and **6.39s** test-time inference. In contrast, Gemini-2.0-Flash and GPT-4o-mini require larger parameters and >22min inference time. These results demonstrate that our method offers a trade-off between performance and computational cost. In addition, it is also worth noting that although MMSLF has a small number of parameters, its GFLOPs are relatively higher. This is because we do not compress the input sequences' length as the prior methods [1, 2]. With further optimization (*e.g.,* sequence dimension reduction), the computational cost of MMSLF can be further reduced.
>
> Table 2. Comparison of Efficiency (Evaluate on SIMS dataset). **Bold** results indicates the optimal result. The experiments were conducted on a Computer with an AMD EPYC 7513 CPU and an NVIDIA Tesla A40 GPU.
>
> |||||||
> | --------------- | -------------- | --------------- | ------------------------------------ | ---------------------- | ----------------------- |
> | **Models**      | **Parameters** | **GFLOPs**      | **Inference Time on SIMS Test Set**  | **F1** **on** **SIMS** | **MAE** **on** **SIMS** |
> | GPT-4V          | >7B            | -               | >30min (affected by network quality) | 81.24                  | -                       |
> | GPT-4o-mini     | >7B            | -               | >27min (affected by network quality) | 82.51                  | 0.453                   |
> | Gemini2.0-Flash | >7B            | -               | >22min (affected by network quality) |  **84.69**| 0.381                   |
> | TFN             | 35.63M         | **0.10** | **3.46s**                                | 77.83±1.62             | 0.434±1.12              |
> | MISA            | 21.66M         | 7.32     | 12.32s                               | 76.54±1.67             | 0.451±1.83              |
> | Self-MM         | **0.38M**      | 6.66     | 11.40s                               | 77.72±0.68             | 0.418±1.05              |
> | TETFN           | 1.53M          | 6.72     | 26.57s                               | 79.34±0.52             | 0.422±1.30              |
> | ALMT            | 2.60M          | 7.00     | 16.08s                               | 80.17±0.60             | 0.421±0.69              |
> | MMSLF-Teacher   | 2.54M          | 96.16    | 12.31s + 27min                       | 84.06±0.43| **0.370±0.50**          |
> | MMSLF-Student   | 0.82M          | 8.61     | 6.39s                            | 81.85±1.41             | 0.382±1.39              |
> |||||||
>
> **Response to Cons 3**
>
> Thank you for the suggestion. We include a visualization in **Appendix Section D.6** and **Appendix Figure 4** that illustrates the alignment learned by the student model under the guidance of MLLMs. The results show that prompts from MLLMs can improve the student’s multimodal alignment. As images and hyperlinks are not supported in the rebuttal, we will include more qualitative cases in the revised version to further highlight the unique contributions of our method.
>
> **Response to Q1**
>
> Please see the **Response to Cons 1**.
>
> **Response to Q2**
>
> Thank you for the question. Considering both cost and performance, we obtain all prompts once before training the teacher model. We also experimented with generating three prompts per sample and randomly sampling one during training on SIMS dataset, but found that this way lead to higher cost without significant performance improvement. **As shown in Table 3 below**, the "obtain once prompt" setting achieves comparable or even better results. Moreover, we observe that the “sampling from three prompts” strategy introduces noticeably higher variance for MMSLF-Teacher, especially in MAE and Corr (e.g., 2.71 vs. 0.50 for MAE std), suggesting that sampling different prompts may introduce inconsistent guidance and lead to unstable training. However, for MMSLF-Student, the variance across runs is relatively small in both settings (e.g., F1 std: 1.41 vs. 0.66), indicating that the student model is less sensitive to prompt sampling. This is likely because the student learns from the teacher’s distilled representations and attention patterns, rather than directly using the prompts.
>
> Table 3. Effect of prompt sampling methods on performance. **Bold** results indicates the optimal result.
>
> | | | | | |
> | ------------------------------------------- | -------------- | -------------- | -------------- | -------------- |
> | Method | Acc-2          | F1             | MAE            | Corr           |
> | MMSLF-Teacher (Sampling from Three Prompts) | **83.57±1.73** | 83.23±1.30     | **0.370±2.71** | 0.682±4.69     |
> | MMSLF-Student (Sampling from Three Prompts) | 81.05±0.66     | 81.18±0.66     | 0.385±0.92     | 0.667±0.87     |
> | MMSLF-Teacher (Obtain Once)                 | 83.06±0.95     | **84.06±0.43** | **0.370±0.50** | **0.690±0.80** |
> | MMSLF-Student (Obtain Once)                 | 81.40±1.58     | 81.85±1.41     | 0.382±1.39     | 0.662±1.26     |
> | | | | | |
>
> **Reference**
>
> [1] MISA: modality-invariant and -specific representations for multimodal sentiment analysis. In ACM MM 2020.
>
> [2] Learning language-guided adaptive hyper-modality representation for multimodal sentiment analysis. In EMNLP 2023.

---

> > ### Comment · Reviewer_RVt6 · 2025-08-05
> > **Response to rebuttal**
> >
> > Thank you for providing a detailed response to each of my questions. Most of my concerns have been addressed in the rebuttal. However, I still believe the paper somewhat overstates its contributions. In particular, the comparison with MMML is not reported. If the paper intends to claim a state-of-the-art (SOTA) result, it should demonstrate superior performance compared to MMML. Without such evidence, the SOTA claim would not be accurate.
> >
> > That said, achieving SOTA performance is not a prerequisite for publication. In my view, the most important aspects are the uniqueness of the insights and the soundness of the experimental support. Therefore, I recommend that the authors revise or qualify the SOTA claim if the results do not surpass those of MMML.
> >
> > Finally, given that most of my concerns are resolved, I will raise my rating.

---

> ### Author Response · Authors · 2025-08-05
>
> **Thank you for your feedback and for raising your score! And sorry for the late of the results of the MMML comparison. We have been working to address all your concerns.**
>
> Consistent with MMML, we replaced the original text encoder and audio encoder with RoBERTa and HuBERT. Then, we ran the code three times on largest MOSEI datasets and report the results in the table below. We can see that the student outperforms MMML on most metrics. We also observed an interesting phenomenon: the student slightly outperforms the teacher on MAE and on some of the Acc-2 metrics. For example, the teacher achieves an MAE of 0.521 while the student achieves an MAE of 0.508. We believe that this occurs because we retained direct supervision from the ground truth labels rather than fully aligning with the teacher during student training, so some of the teacher’s subtle biases were not transferred.
>
> We sincerely appreciate your suggestions and have agree your opinion that the findings and the experimental support are important. What we want to emphasize is also these points. In the revised manuscript, we will highlight these points and include MMML comparisons across the datasets.
>
> Table 3. Comparison with MMML on MOSEI dataset. Bold results indicates the optimal result.
> |     |                           |                           |                |                |
> | -------------------------- | ------------------------- | ------------------------- | -------------- | -------------- |
> | Method                     | Acc-2                     | F1                        | MAE            | Corr           |
> | MMML | **86.32**/86.73           | 86.23/86.49           | 0.517          | 0.791          |
> | MMSLF-Teacher (Three Runs) | 85.66±0.63/**87.70±0.10** | 86.08±0.37/**87.85±0.04** | 0.521±0.16     | **0.800±1.04** |
> | MMSLF-Student (Three Runs) | 86.08±0.32/86.98±0.02     | **86.29±0.20**/87.03±0.04     | **0.508±0.24** | 0.797±0.24     |
> |||||

---

> > ### Comment · Reviewer_RVt6 · 2025-08-06
> > **Official Comments by Reviewer**
> >
> > Thanks for the reply. I am curious about the audio encoder. The MMML uses Data2Vector in its main results. I think you should compare it with this encoder, rather than HuBERT. If you can obtain a better result using Data2Vector, you can also claim the SOTA. If not, please remove the SOTA in this paper.

---

> > > ### Author Response · Authors · 2025-08-09
> > >
> > > Dear Reviewer RVt6,
> > >
> > > **Thank you very much for your time and for engaging in multiple rounds of discussion regarding our paper.** We believe that your suggestions will further improve the quality of our work. The relevant content from our discussions will be revised into the paper.
> > >
> > > Best Regards,
> > >
> > > The Authors

---

> ### Author Response · Authors · 2025-08-07
>
> Thank you for your response and for your engagement in the discussion. The MMML paper mentions that different encoders (i.e., Data2Vec and HuBERT) were used for different datasets to obtain optimized extraction of features, thus achieving a solid foundation for the subsequent fusion process. And according to your suggestion, we also conducted experiments using RoBERTa and Data2Vec as the text and audio encoders, respectively. The results are shown in the Table below. Although MMML achieves higher performance on Acc2-Has0 (86.32%) and F1-Has0 (86.23%), our method outperforms on more metrics. For example, the teacher achieves better results on Corr (0.792±0.15). The student achieves better results on Acc2-Non0 (87.09±0.25), F1-Non0 (87.18±0.24) and MAE (0.513±1.27). Interestingly, similar to the case when using HuBERT as the encoder, the student model slightly surpasses the teacher on some metrics. We believe that ground-truth supervision was retained alongside the teacher signal in student training allowing the itself to avoid some potential noise or biases learned by the teacher.
>
> **After referring to your suggestions and having a discussion among the authors, we think a cautious description would be more appropriate when MMML is added for comparison. In the revised version, we will revise the use of the term “SOTA” and opt for a more accurate description of our results. For example, explicitly state on which metrics the model achieves the better performance, such as Acc2-Non0, F1-Non0, MAE, and Corr.**
>
> Finally, we consider the comparison with MMML to also serve as an discussion that shows the impact of different encoder choices on our model performance. We will include further discussion in the Appendix.
>
> **We sincerely appreciate your feedback and hope that our response can address your concerns.**
>
>
> Table 5. Comparison with MMML on the MOSEI dataset. RoBERTa and Data2Vec are used as the text and audio feature extraction, respectively. Bold values denote the best results for each metric.
>
> |                  |                           |                           |                |                |
> | -------------------------- | ------------------------- | ------------------------- | -------------- | -------------- |
> | Method                     | Acc-2                     | F1                        | MAE            | Corr           |
> | MMML                       | **86.32**/86.73           | **86.23**/86.49           | 0.517          | 0.791          |
> | MMSLF-Teacher (Three Runs) | 85.47±0.25/87.06±0.46 | 85.53±0.27/87.16±0.38 | 0.522±1.31     | **0.792±0.15** |
> | MMSLF-Student (Three Runs) | 85.93±0.58/**87.09±0.25**     | 86.08±0.55/**87.18±0.24**     | **0.513±1.27** | 0.785±1.66     |
> |                  |                           |                           |                |                |

---

### Author Response · Authors · 2025-08-09

Dear Reviewers and ACs,

**We would like to thank all the reviewers for their valuable feedback, and thank the AC for their behind-the-scenes efforts.** We are encouraged that you found our motivation/insights/framework interesting/clear/effective (Reviewers RVt6, EAd9, ctoX, Wv59). We appreciate your recognition of our clear and well-organized presentation (Reviewers RVt6, Wv59), technical correctness (Reviewers EAd9), and persuasive results (Reviewer EAd9). We are glad that Reviewers EAd9 and Wv59 believe the framework, which leverages MLLM knowledge for conditional alignment while keeping the student model cost-effective at inference, to be compelling. We also thank that all the reviewers noted the additional experiments/discussions/clarifications provided during the rebuttal can further strengthen the paper (Reviewers RVt6, EAd9, ctoX, Wv59)!

**We believe all your suggestions have played an important role in improving the quality of our paper.** Specifically, for additional experimental results, we have:

- provided comparisons with more works including  MMML, TCAN, DeepMLF, MAG-BERT, DMD and DLF (see response to cons 1 and Q1 of Reviewer RVt6, Response to W2 of Reviewer EAd9);
- included statistical significance analysis (see Response to W2 and L3 of Reviewer ctoX);
- expanded efficiency analysis with concrete parameter, FLOPs, and inference time comparisons (see Response to cons 2 of Reviewer RVt6, Response to Q3 of Reviewer EAd9, Response to W1 and L1 of Reviewer ctoX, Response to W2 of Reviewer Wv59);
- discussed the effect of prompt sampling methods on performance (see Response to Q2 of Reviewer RVt6);
- discussed the framework’s sensitivity and robustness to different MLLM prompt templates  (see Response to Q2 of Reviewer EAd9);
- discussed whether the framework can serve as a plug-in for other methods (Response to W7 and Q1 of Reviewer EAd9).

**For areas that all you indicated require further clarification**, we have:

- revised the use of the term “SOTA” and adopted a more accurate description of our results, explicitly stating the metrics where the model achieves better performance (see the discussion history with Reviewer RVt6);
- improved clarity in the abstract, introduction, and conclusion to clearly present the main idea and highlight the innovativeness of our framework (see Response to W1 and W3 of Reviewer EAd9);
- improved the captions of tables to provide more detailed explanations (see Response to W4 of Reviewer ctoX);
- clarified why Gemini-2.0-Flash underperforms GPT-4o-mini within our framework despite showing comparable or better raw performance when used directly for the task (see Response to W3 of Reviewer ctoX, Response to W3 and Q1 of Reviewer ctoX);
- clarified the differences in some metrics between Gemini-2.0-Flash and GPT-4o-mini (see Response to Q4 of Reviewer Wv59);
- clarified the reason for the performance drop without MLLM prompts and why it stems from architectural constraints (see Response to W6 of Reviewer EAd9);
- explained the existing visualization and clarified that more will be added in the revised paper (see response to cons 3 of Reviewer RVt6);
- clarified the difference between regression and classification in MSA, and described the procedure for converting regression outputs into sentiment labels for F1 and Accuracy computation (see Response to Q2 of Reviewer Wv59);
- clarified the motivation behind the model design (see Response to W4 and W5 of Reviewer EAd9, Response to W3 of Reviewer Wv59);
- clarified framework's relevance with other multimodal tasks (see Response to Q4 of Reviewer EAd9) and clarified the details of some techniques/baseline models (Response to L2 of Reviewer ctoX, Response to W1, Q1, and Q3 of Reviewer Wv59).

**Finally, we sincerely thank you for your time, effort and positive feedback. We believe that all your feedback is important in further improving the quality of our paper.**

Best Regards,

The Authors

---

### Decision · Program_Chairs · 2025-09-17

**Decision:**

Accept (poster)

**Comment:**

The paper introduces a teacher-student framework for multimodal sentiment analysis (MSA) that leverages multimodal large language models (MLLMs) to guide the training of efficient, task-specific models. Reviewers generally agree that the motivation is timely and the approach is novel, with comprehensive experiments and clear methodology.

The framework demonstrates competitive performance and significant efficiency gains, especially in the student model, which is much less resource-intensive than direct MLLM inference. However, reviewers caution against overstating state-of-the-art (SOTA) claims, noting that the method does not consistently outperform all recent baselines across every metric.

They also highlight the need for clearer discussion of the trade-off between performance and efficiency, and suggest improvements in presentation and statistical analysis.

The authors responded thoroughly to these concerns, providing additional experiments, fairer comparisons, and statistical significance testing, and committed to further clarifying claims and improving the manuscript. Overall, the work is seen as a solid and practical contribution to efficient multimodal sentiment analysis, with most reviewer concerns satisfactorily addressed.